ecology/bioenergetics

northern gannet, isotope ecology, movement ecology, bioenergetics, accelerometry

**Author for correspondence:**
Ashley Bennison
e-mail: ashley.bennison@ucc.ie

# A bioenergetics approach to understanding sex differences in the foraging behaviour of a sexually monomorphic species

Ashley Bennison[1,2], Joan Giménez[1,3], John L. Quinn[2], Jonathan A. Green[4] and Mark Jessopp[1,2]

[1]Centre for Marine Renewable Energy, and [2]School of Biological, Earth and Environmental Sciences, College of Science, Engineering and Food Science, University College Cork, Ireland
[3]Marine Renewable Resources Department, Institute of Marine Science (ICM-CSIC), Barcelona, Spain
[4]School of Environmental Sciences, University of Liverpool, Liverpool L69 3GP, UK

AB, 0000-0001-9713-8310; MJ, 0000-0002-2692-3730

Many animals show sexually divergent foraging behaviours reflecting different physiological constraints or energetic needs. We used a bioenergetics approach to examine sex differences in foraging behaviour of the sexually monomorphic northern gannet. We derived a relationship between dynamic body acceleration and energy expenditure to quantify the energetic cost of prey capture attempts (plunge dives). Fourteen gannets were tracked using GPS, time depth recorders (TDR) and accelerometers. All plunge dives in a foraging trip represented less than 4% of total energy expenditure, with no significant sex differences in expenditure. Despite females undertaking significantly more dives than males, this low energetic cost resulted in no sex differences in overall energy expenditure across a foraging trip. Bayesian stable isotope mixing models based on blood samples highlighted sex differences in diet; however, calorific intake from successful prey capture was estimated to be similar between sexes. Females experienced 10.28% higher energy demands, primarily due to unequal chick provisioning. Estimates show a minimum of 19% of dives have to be successful for females to meet their daily energy requirements, and 26% for males. Our analyses suggest northern gannets show sex differences in foraging behaviour primarily related to dive rate and success rather than the energetic cost of foraging or energetic content of prey.

# 1. Introduction

Many animals show sex-specific foraging differences, though it is often difficult to explore the mechanisms behind these differences—particularly in free-ranging predators. Sex differences in foraging are often pronounced in sexually dimorphic species [1,2]. These differences may be due to competitive exclusion [3], or where different sexes may have access to different foraging areas due to their size [3,4] or foraging habitat preference [5]. Divergent sexual behaviours may also represent differences in nutrient requirements or prey preferences [6,7], levels of parental care [8] or in the energetic demands of locomotion [9,10]. Differences can also arise because a dominant sex will outcompete or displace the other, resulting in sexual segregation [11,12], niche expansion and reduced intraspecific competition [13]. For example: giant petrels, *Macronectes giganteus*, where females weigh 80% the mass of males, show spatially segregated foraging areas [14], a pattern that holds true across a wide variety of taxa [15–18]. Although sex differences in foraging tend to be less obvious in sexually monomorphic species, they still occur [19].

In monomorphic species, sex-specific foraging behaviours can be driven by differing energy requirements between the sexes [20]. Foraging in different locations will provide different resources that may be required in different amounts between the sexes. For example, Barau's petrel (*Pterodroma baraui*) is a monomorphic seabird where males and females forage in different locations early in the breeding season, as females must restore body condition after egg production [20]. There is also evidence to suggest that sex-specific foraging strategies in sexually monomorphic species may be driven by intraspecific competition causing one sex to be displaced spatially or to forage in different niches [21]. For example, brown boobies (*Sula leucogaster*) are considered to show sex differences in foraging, as competition for resources suggests that males exclude females from foraging on squid, and this exclusion may change with different levels of foraging resources available. [22] Foraging theory states that animals attempt to intake food in the most optimal manner possible [23–25] to ensure that net energy gain exceeds gross energy expenditure. However, accurately measuring energy intake and expenditure remains a challenge, especially in free-ranging animals [26,27].

Measuring energetic expenditure has previously involved the use of double-labelled water, respirometry chambers or heart rate loggers [28]. Though heart loggers can be used to investigate behaviour-specific energy costs [29] and respirometers can provide resting metabolic rates and calibration for other field measurements [30], these techniques can be invasive. In recent years, accelerometry studies on free-ranging individuals have explored energetic expenditure at a much finer scale [31]. These studies can use measures of dynamic body acceleration (DBA) as a proxy for energy expenditure, due to a strong correlation with the volume of oxygen consumed by muscles during a given time period ($VO_2$) [32–34]. However, developing a complete understanding of how accelerometry signals relate to energy use and the corresponding energy budgets of an individual animal requires knowledge of diet and energetic intake.

Net energy intake is determined by the energy gained from successful foraging against the energy expended for basal metabolism and for activities such as locomotion. Quantifying energy gained through diet in free-ranging animals can be difficult without invasive techniques such as stomach content analysis [35] or direct observation of prey capture [36]. However, stable isotope analysis (SIA) is a minimally invasive technique that can provide diet information and, in seabird studies, is known to correlate well with these other more direct methods such as regurgitate sampling and direct observation of foraging [37–39]. Isotopic ratios of $^{12}C/^{13}C$ and $^{14}N/^{15}N$ can be used to infer prey species consumed by an individual [40]. Both carbon and nitrogen can be considered as indicators of the trophic level an animal is foraging at [41]. Nitrogen isotopes enrich at a faster rate in predators than carbon isotopes, but the ratio between them can inform trophic level, trophic niche width and diet [42]. Using SIA to predict predator diet can therefore provide insight into the energetics of foraging.

The northern gannet (*Morus bassanus*), hereafter gannet, is sexually monomorphic with no significant morphological differences between adult males and females [43,44]. While females are marginally heavier than males on average, (average 200 g—approximately 6% difference [45]), there is considerable overlap, and mass alone cannot be used to sex individuals [45]. Despite the lack of overt sexual dimorphism, all populations of the species studied thus far show strong sexually divergent foraging strategies. Female gannets are more selective in choosing foraging grounds [46] and undertake longer trips, further offshore than males, a pattern that is thought to arise from habitat segregation [47]. From a dietary perspective, male gannets consume higher proportions of fisheries discards than females, a division thought to derive from the competitive exclusion of female gannets

from vessels [48], and is a distinction only present in breeding adults [44]. Females which specialize on fisheries discards travel shorter distances than females which specialize on forage fish; however, this distinction is not apparent among males [49]. At present, there is no clear evidence for whether male and female gannets target different-sized prey items. A lack of strong sexual dimorphism in gannets suggests that sex differences in foraging strategies and diet may derive from different energetic demands between the sexes caused by differential responsibilities during chick rearing [50], a previously untested hypothesis.

In the present study, we used GPS, accelerometry and SIA data to gain a better understanding of how gannets engage in foraging and how different demands upon the sexes may affect foraging strategies. Specifically, we explore sex differences in foraging of gannets in terms of diet, dive types, frequency of prey capture attempts and the energetic cost of prey capture attempts. Additionally, we quantify the energetic requirements of each sex, taking into account energy expended during foraging and, using data from published studies, energetic demands of feeding offspring. Finally, we consider minimum dive success rates necessary for male and female gannets to meet their energy demands.

Specifically we aim to test the following hypotheses:

(1) Sex differences in the foraging ecology of gannets derive from the different energetic demands placed upon the sexes by differential responsibilities during chick rearing.
(2) Due to limited sexual dimorphism, there is no difference in the cost of similar prey capture attempts between the sexes.
(3) Due to differing energy demands and foraging behaviour, the sexes have different prey capture success rates.

## 2. Methods

All data collected as part of this study are available from the Dryad Digital Repository: https://doi.org/10.5061/dryad.zs7h44j88 [51].

### 2.1. Data collection

A visual diagram of the methodology is presented in figure 1. Breeding adult gannets ($n = 8$ in 2017, $n = 6$ in 2018) attending three- to five-week-old chicks were tracked from Great Saltee, southeast Ireland (52° 7′37.92″ N, 6°35′45.6″ W). In 2017, three female, four male and one unknown gannets were tagged; four males and two females were then tagged in 2018. Birds were equipped with tags for an average of 3.70 ±1.39 days. To reduce potential impact on a breeding pair, only one individual of a pair was tagged for this study. Birds were caught using an 8–10 m pole with a metal crook, weighed and equipped with a combination of dataloggers. GPS loggers (i-gotU GT-120, Mobile Action Technology Inc., Taipei, Taiwan, 14 g, dimensions: $4 \times 2 \times 1$ cm) recorded locations every 3 min; time depth recorders (TDR, CEFAS G5, 2.5 g, dimensions: $2 \times 1 \times 1$ cm) recorded depth at 4 Hz after exceeding depth threshold of either 0.5 m or 1 m depending upon tag set-up; tri-axial accelerometers (Gulf Coast Data Concepts X16-mini, 17 g, dimensions: $6 \times 2 \times 1$ cm) recorded $g$-forces (1 $g = 9.807$ m s$^{-2}$) at 50 Hz. GPS and TDR loggers were attached ventrally to two–three central tail feathers using strips of waterproof Tesa tape. Accelerometers were attached to 10–15 mantle feathers between the wings. Three birds in 2017 and six birds in 2018 were equipped with GPS, TDR and accelerometers, while the remaining birds were equipped with only GPS and accelerometers. Total instrument mass was less than 2% of body mass and positioned to minimize impact on gannet movement, both aerodynamic and hydrodynamic [52]. It is important to consider effect of tag attachment [53]. Previous studies of gannets have employed similar devices [48,54,55]; the relatively small load placed upon gannets during these studies means that it is unlikely that gannet behaviour would be impacted from tag attachment of this magnitude. A table of deployment weights can be seen in electronic supplementary material, table S1, showing that many gannets gained weight or lost very small amounts during the tracking period. Blood samples were taken from 47 birds ($n = 19$ in 2017 and $n = 28$ in 2018), including the accelerometer-equipped birds, and used to construct a population model of dietary intake from isotope analysis (see section ‘*Isotopic analysis for diet composition*’ below). Between 1 and 1.5 ml of blood was sampled from the tarsus vein for SIA (see below), and two–three breast feathers were plucked for genetic sexing following the method outlined by Griffiths *et al.* [56]. Though we do not have data on reproductive success, all pairs were observed to continue in normal chick feeding behaviour, and chicks were observed to be alert and healthy, during and after the study period.

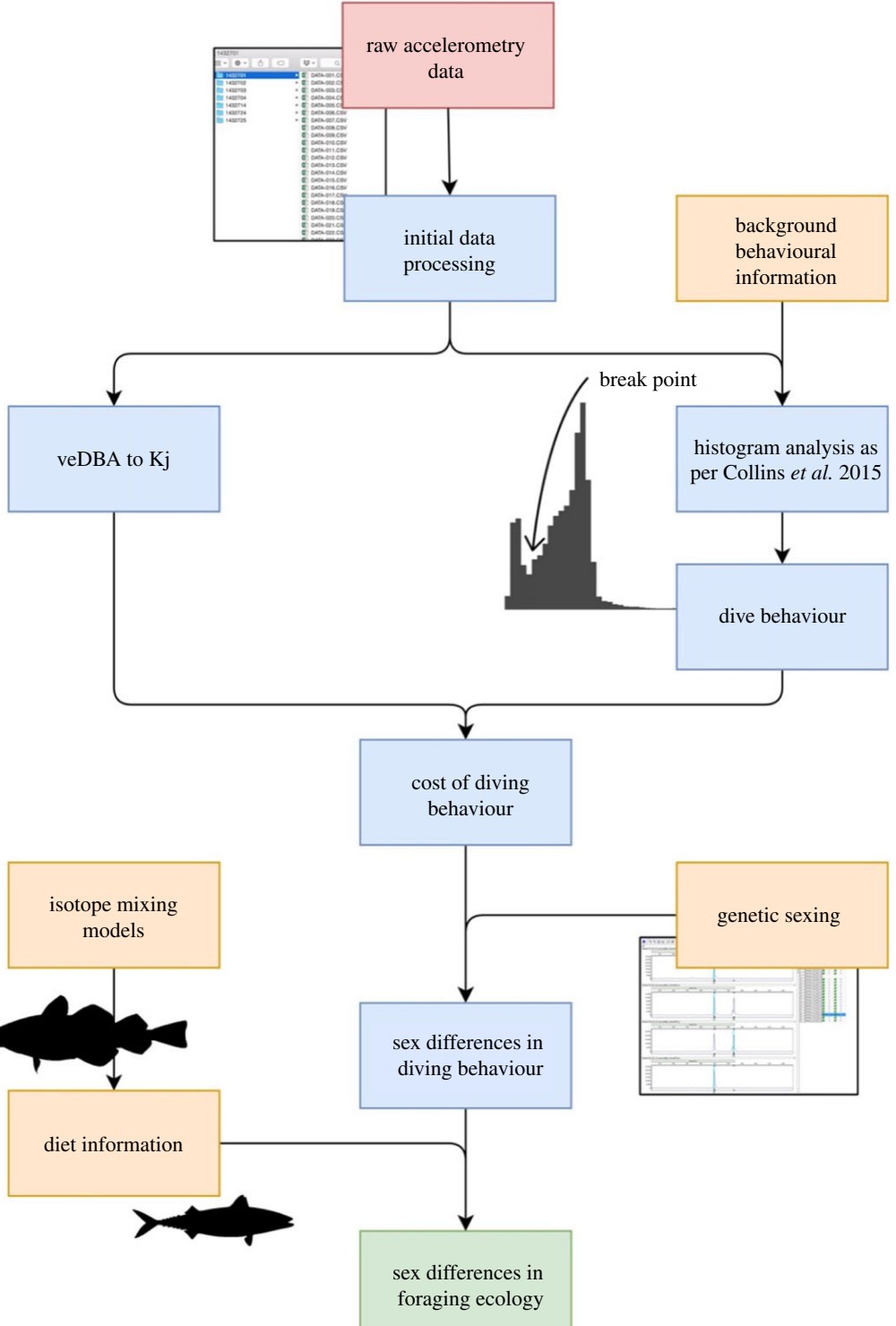

**Figure 1.** Conceptual diagram of a methodology for data processing and the steps required to explore the sex differences in the foraging of northern gannets. The process starts at top with the red box labelled 'raw accelerometry data' and ends with the green box 'sex differences in foraging ecology'. Blue boxes represent the methodology for analysing data and orange boxes represent additional analysis.

## 2.2. Data processing and dive behaviour definition

Behaviour classification from accelerometry data used a thresholding approach. Thresholds were determined using protocols and guidance set out by Collins *et al.* [57] and Shepard *et al.* [58]. Diving

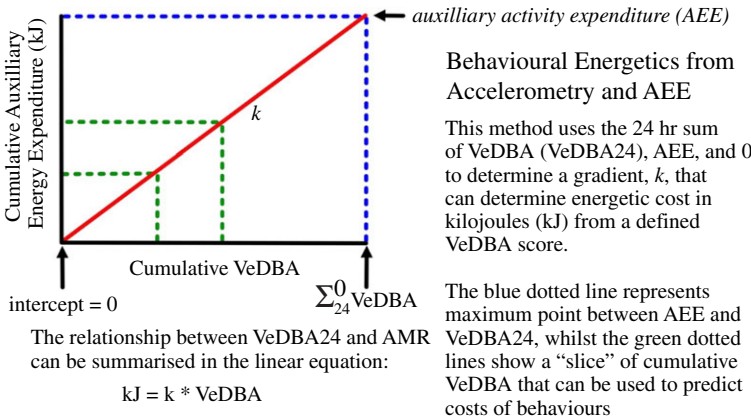

## Estimating Energetics of Movement from Accelerometry

for energy consumed only by movement:

← *auxilliary activity expenditure (AEE)*

### Behavioural Energetics from Accelerometry and AEE

This method uses the 24 hr sum of VeDBA (VeDBA24), AEE, and 0 to determine a gradient, *k*, that can determine energetic cost in kilojoules (kJ) from a defined VeDBA score.

The blue dotted line represents maximum point between AEE and VeDBA24, whilst the green dotted lines show a "slice" of cumulative VeDBA that can be used to predict costs of behaviours

Cumulative Auxilliary Energy Expenditure (kJ)

*k*

Cumulative VeDBA

$\Sigma^{0}_{24}$VeDBA

intercept = 0

The relationship between VeDBA24 and AMR can be summarised in the linear equation:

kJ = k * VeDBA

**Figure 2.** Conceptual diagram demonstrating how to estimate the AEE (in kJ) for a given quantity of VeDBA units within a dataset where it is assumed that energy above basal metabolism is consumed only by movement.

events occurred when average acceleration (running average of 2 s) in the X-axis (also known as the surge axis) was less than $0g$ and standard deviation (s.d.) in the mean X-axis was greater than $1.4g$. The end of a dive was defined by a 1 second lagged maximum of pitch change within a 60 s period from the start of a dive. Take-off events were defined with a threshold where, following a dive, the s.d. of the Z-axis (also known as the heave axis) was greater than $1.8g$ and the s.d. of the X-axis was greater than $1g$. Take-off events were considered to have ended and returned to normal flight when the s.d. of the Z-axis resolved to less than $1.4g$ and the s.d. of the X-axis was less than $1.4g$. Data from a subset of birds ($n = 9$) tagged with both TDRs and accelerometers were used to validate accelerometer-derived dive events by visually comparing timestamps to TDR-confirmed dives; this required each dive to be manually viewed and checked to compare with a dive from a TDR and all dives from between accelerometry and TDRs were matched successfully. Accelerometer-derived dives had a bimodal distribution and were split into plunge dives and pursuit dives based on a distinct break within the frequency distribution at 5 s (see electronic supplementary material, figure S1); plunge dives are dives followed by an almost immediate rise to the surface, while a pursuit dive is characterized by sustained chase of prey underwater.

## 2.3. Energetics from accelerometry

DBA is a relative metric that can be used as a proxy for energetic expenditure from animal movement [59] and can be used to develop highly accurate activity budgets [60]. We used vectorial DBA (VeDBA) to account for any variation in tag alignment [61]. VeDBA was calculated for every second within the tracking period. The best practice to estimate energetic expenditure from DBA is to have species and behaviour-specific relationships between the rates of these quantities [62]. However, such calibration relationships do not exist for the overwhelming majority of species, requiring an alternative approach to be adopted that we outline here. This approach is based on the observation that the relationship between the rate of energy expenditure (kJ) and instantaneous VeDBA is linear among a variety of animal taxa, including mammals, reptiles and birds [26,33,62], with slope *k*. Using published allometric estimates of energy expenditure, it is possible to produce estimates of kilojoules expended in the movement for given values of VeDBA, via the process outlined in figure 2. The basis of the process is the assumption that energy expended in movement is equal to an animal's field metabolic rate (FMR) minus basal metabolic rate (BMR) for any given period; also known as the activity metabolic rate, auxiliary energy expenditure (AEE) or daily energetic scope [63,64]; here, we use AEE. AEE is then the energy expended, in kilojoules, for a given 24 h period in movement alone. The total sum of VeDBA over a 24 h period (VeDBA24) is then equivalent in energy to the AEE and we assume that where VeDBA is equal to 0, then energy expended in locomotion must equal 0 kJ, which would be the case for a completely inactive animal. We can then construct a simple relationship between AEE, VeDBA24 and the origin (where energy and movement are both 0) that allows the prediction of kilojoules expended from each unit of VeDBA expended in that day. Here, we used FMR estimates for northern gannets provided by the seabird FMR calculator [65,66], corrected for individual bird mass

and colony latitude (52° N), and BMR estimates provided by allometric equations from Schreiber and Burger [67] to produce VeDBA to kJ gradients for each individual. VeDBA24 was calculated for each complete 24 h period for each bird (range 1–5) and a mean value calculated per bird, which was used in the predictive equation along with individual-specific estimates of BMR and FMR from that bird. This allowed for incorporation of an individual's mass which improved precision and allowed an estimate of the gradient between VeDBA and energy expenditure (constant $k$ in figure 2, but also electronic supplementary material, figure S2 for individual gradients) to be estimated for each individual. To assess the effectiveness of the calculations underlying this approach, we used the individual relationships produced in this methodology to predict AEE from VeDBA24 for each bird. As this was successful (see electronic supplementary material, table S2), we accepted the use of this methodology. It is important to note that this method only allows for the prediction of energy expended by movement and assumes that all acceleration is due to movement; at present, it is not possible to effectively account for incidental records of acceleration not due to movement, such as periods of rest on water where sea swell may be detected by the accelerometer. However, if future work could filter out such incidental acceleration from acceleration due to movement, then the methodology would be further enhanced. Furthermore, the approach assumes BMR and FMR to be constant throughout the tracking period and driven by mass (and latitude in the case of FMR) alone. This ignores likely inter-individual variation in both FMR and BMR as well as any sex-specific differences that might exist in these quantities other than those accounted for by mass. However, since the relevant allometric equations do not incorporate sex effects, they are most effective for predicting population-level estimates for equal proportions of males and females. For that reason, the approach is only applicable for groups of individuals, including single-sex groups, rather than individual rates of energy expenditure, as is the case in the present study.

We used the individual gradient between VeDBA and energy expenditure shown in figure 2 to estimate total energetic *expenditure* for an individual bird from the time it left the colony, to the point of recapture. Gannet trips may range from one to several days, and so this approach allowed our predictions to account for a full range of behaviours, from in colony, transiting and foraging. This also allowed for energy expenditure to exceed FMR, which would be the case for a bird which spends more time resting than the 'average' bird used in population-level estimates of FMR used to build allometric relationships. As gannet foraging trips may last several days, they incur increasing energetic costs during a foraging trip, such as feeding chicks upon return. We have included this in the analysis by considering energetic differences in individual AEE per 24 h period. We then calculated total energetic *demands* (TED) by adding to this value the energetic demand of raising a four-week-old chick of 1397.14 kJ day$^{-1}$ [50]. Due to unequal parenting roles, this cost was split with females retaining 60% of this cost and males 40% [50,68]. For multi-day trips, chick demands were multiplied by the appropriate length of time for energetic demand, and then also including BMR estimates for each 24 h period. Though it would be most appropriate to have information on feeding rates of chicks in this study, we do not have this information and instead consider the overall energy requirements of chicks which act as a proxy to feeding rates. This produced a value of TED for each gannet for each complete 24-hour period (range 1–5 days).

To calculate the energetic cost of dives, VeDBA was summed over the time frame of a dive (from initiation of the dive to completion of the subsequent take-off) and multiplied by that individual's value of $k$ (figure 2) to estimate energy expended in kilojoules for each dive event for each gannet.

## 2.4. Isotopic analysis for diet composition

Blood samples taken during tag deployment were centrifuged for 10 min to separate red blood cells (RBC) from plasma. While RBC therefore represent diet prior to the deployment, preliminary sampling showed that isotopic signatures do not differ significantly between blood samples collected on deployment and recovery of devices approximately one week apart (unpublished data). Stable isotope analyses were performed at Elemtex UK (Stable Isotope and & Elemental Analysis Expertise), using a Thermoquest EA1110 Elemental Analyser linked to a Sercon 2020 stable isotope ratio mass spectrometer running in continuous flow mode. Accuracy and precision were monitored through laboratory internal standards and an in-house comparison standard nested within samples.

Prey stable isotope values were obtained from a published dataset of Celtic Sea fish samples [69]. Jennings & Cogan [69] conducted SIA of samples without lipid extraction; therefore, the $\delta^{13}$C data included in the published dataset are not corrected for differences in lipid content, but the % C and N data was used to make the required corrections following Logan *et al.* [70]. As recommended by

Phillips *et al.* [71], a reduced prey dataset was used and included only those species previously recorded in more than 3% of the diet for Great Saltee gannets [72]. These species can be seen in electronic supplementary material, table S3.

Using Bayesian isotopic mixed models, it was possible to compare blood values to reference prey values to reconstruct the diet of gannets. The model was run on 'long' settings (chains = 3, length = 300 000, burn-in = 2 000 000, thinning = 100), using average diet-to-tissue discrimination factors (2.25 ± 0.61‰ for $\delta^{15}$N and 0.24 ± 0.79‰ for $\delta^{13}$C) from various studies of piscivorous birds [73–76]. Model convergence was assessed with the Gelman–Rubin diagnostic [77]. Sex-based diet estimates were obtained through Bayesian mixing models using the R package 'MixSIAR' [78]. We fit several models of diet with fixed and random effects as covariates and evaluated the relative support for each model using LOO (leave-one-out cross-validation) weights [79]. Model outputs were then used to construct prey proportions in the diet of males and females in 2017 and 2018.

We assumed the sizes of individual prey species were similar to those in Lewis *et al.* [72], a study from the same colony that did not identify any difference in the size of fish caught between the sexes. The size and mass of the fish were then used to calculate the energetic content (in kJ) of each fish species (using allometric equations referenced by Lewis *et al.* [72] and assuming a 76.1% assimilation efficiency following Cooper [80], see electronic supplementary material, table S3). For each sex-specific diet, the energy content (kJ) of each fish was multiplied by the proportions of species in the diet, and these proportional values were summed to provide an average kJ intake value (KIV) for a successful dive (a dive resulting in prey capture) for each individual gannet, assuming a successful dive results in the capture of one prey item.

## 2.5. Statistical analysis

A Mann–Whitney–Wilcoxon test was used to test weight differences between the sexes. An unpaired *t*-test was also undertaken to test the differences in dive length between males and females. To explore sex differences in the overall cost of prey capture attempts (dive and subsequent take-off), a linear mixed effect regression (LMER) was used to test for sex differences in dive and take-off characteristics. Factors included year, sex, mass, dive type and the interaction between sex and dive type with energy expenditure (kJ) as the dependent variable. Individual was included as a random effect to account for repeated measures of the same individual. The interaction between sex and dive type was included to explore if the different masses of the sexes (approximately 200 g [45]) impacted the cost of a dive type. To select the most parsimonious model, the dredge function from the 'MuMIn' package was used [81]. Using the model averaging function in the MuMIn package, any models within six AIC values were kept and model averaging undertaken [82]. A difference in dive rate (dives per day) between females and males was tested using a general linear model using the dive rate as a response variable for each individual and sex as a predicting factor. To determine if sex influences AEE plus chick energetic demands, an LMER was used to predict AEE (per day) from sex and year, with ID as a random effect to account for repeated measures from individuals.

A Mann–Whitney–Wilcoxon test was used to test for differences in KIV between sexes. For each gannet, TED was divided by KIV to determine how many successful dives were required to maintain body condition, forage and provision for a chick, assuming no change in body mass. The number of successful dives required was then considered as a proportion of the number of dives undertaken, therefore presenting a minimum percentage of dives which must have been successful for each individual gannet to maintain body mass and conduct its role in chick provisioning.

# 3. Results

Of the 14 gannets tracked, five were female, and eight were male. The DNA test of one individual was inconclusive and so was recorded as unknown sex; this individual was not included in the analysis of sex differences. Male gannets were on average lighter than females; male mass was 2.70 kg ±0.19 with females weighing 2.99 kg ±0.15 (Wilcoxon test: $W$ = 35.5, r = 0.88, $p$ = 0.025).

## 3.1. Sex differences in dive behaviour

We detected 1046 visually validated dives and subsequent take-off events. Of these dives, 24% were pursuit dives with females having a slight tendency towards pursuit dives compared with males.

**Table 1.** Conditional model summary from the averaged mixed effect linear regression used to predict kilojoules (kJ) expended during a prey capture attempt. Input variables were year (2017 and 2018), sex (male and female), dive type (pursuit or plunge) and mass. The interaction between sex and dive type was also included. Dive type (plunge) and sex (female) were absorbed into the intercept.

| dive energetics model | coefficient | s.e. | adjusted s.e. | z-value | p-value |
|---|---|---|---|---|---|
| intercept | −1.334 | 1.485 | 1.487 | 0.898 | 0.369 |
| type (pursuit) | 0.537 | 0.0429 | 0.0429 | 12.517 | <0.001 |
| year (2018) | 0.915 | 0.303 | 0.303 | 3.020 | <0.01 |
| mass | 0.167 | 0.782 | 0.783 | 0.214 | 0.8306 |
| sex (male) | −0.0823 | 0.375 | 0.375 | 0.219 | 0.8264 |

Female dives were 5.19 ± 3.81 s long, and male dives were 5.04 ± 3.53 s long; an unpaired $t$-test confirmed no significant differences in dive length between male and female dives ($t = -0.53$, d.f. = 496.78, $p = 0.59$). Combined cost of a single prey capture attempt (dive + take-off) in females was 1.94 ± 0 0.65 kJ s.d., while for males, it was 1.74 ± 0.83 kJ s.d., suggesting that male dives are 11.2% less costly than female dives. An averaged LMER indicated a significant effect of dive type and year on energy expenditure associated with dives, while sex was retained as a non-significant factor (table 1, and see electronic supplementary material, table S4 for model averaging table). The estimates of the total cost of all prey capture attempts represent less than 4% of the daily total energy expenditure for each individual (electronic supplementary material, table S5). Accounting for unequal provisioning of the chick, and the cost of foraging, daily energetic demands were 10.28% higher for females than males (female TED = 4209 kJ ±110.48 s.d.; male TED = 3817 kJ ±256.78 s.d., Wilcoxon test: $W = 6$, $p < 0.05$, total number of female days: 14.84, total number of male days: 31.88).

The daily dive rate of females was significantly greater than that of males (25.9 and 17.3, respectively, GLM $F_{13} = 8.63$, $p < 0.01$). However, because the cost of individual prey capture attempts is so low, a LMER predicting the AEE (kJ) per day for each individual from sex and year, with ID as a random effect, found no significant effect of sex on AEE (LMER $F_{38} = 0.0018$, $p = 0.96$)

## 3.2. Isotopic analysis

The isotope mixing model predicted that the most consumed prey species were Atlantic mackerel (*Scomber scombrus*) (51.07% ±4.34 s.d.) and European sprat (*Sprattus sprattus*) (9.42% ±4.93 s.d.) followed by lesser sandeel (*Ammodytes marinus*) (8.82% ±4.99 s.d.) and Atlantic herring (*Clupea harengus*) (4.77% ±1.98 s.d.). The remaining species included in the models were each predicted to contribute less than 8% to the overall diet. Seven different models were tested (table 2) and the best model included *Year* as a covariate (model weight: 76.8%, model 4). The second-best model included *Sex* and *Year* as variables with a relative weight of 23.1% and was used to predict sex-specific diets in each study year. There was no support for a model using individual ID only. Diet between the sexes was similar in both years (table 3), though mackerel made a higher contribution to male diet (difference of 3.4% in 2017 and 4.3% in 2018).

Applying average energy content of prey in proportion to its occurrence in the diet, a successful dive was estimated to have a KIV of 1006 kJ for females and 1005 kJ for males in 2017. In 2018, this figure rose with changing diet to 1563 kJ for females and 1553 kJ for males.

Based on the number of dives performed and average energy content of prey in proportion to their occurrence in sex-specific diets, female minimum feeding success rate was calculated as 19.39% ±7.71 s.d., while the male rate was 26.60% ±13.81 s.d. (figure 3). A summary of all results including dives, energy expenditure and success rates can be seen in table 4.

## 4. Discussion

Here we show that, for gannets, sex differences in foraging behaviour are not the result of divergent energetic costs of foraging or different energetic content of consumed prey. We instead suggest that sex differences in foraging behaviour are likely to have arisen from unequal energetic demands

**Table 2.** Bayesian mixed effect model outputs to determine predictors of diet in northern gannets. The best model lent support for a Year only model; however, the second-best model was Sex + Year with a model weight of 23.1%. This model was used to predict diet of the sexes. Leave-one-out cross-validation information criteria (LOOic) were used to assessed model suitability.

| model | variables | LOOic | standard error LOOic | delta LOOic | standard error delta LOOic | weight |
|---|---|---|---|---|---|---|
| 4 | Year | 87.5 | 11.8 | 0 | NA | 0.768 |
| 6 | Sex + Year | 89.9 | 11.6 | 2.4 | 3 | 0.231 |
| 5 | Year (by ID) | 106.8 | 8.6 | 19.3 | 6.4 | 0 |
| 2 | Sex | 109.7 | 10.9 | 22.2 | 6 | 0 |
| 1 | Null | 110.7 | 11 | 23.2 | 5.5 | 0 |
| 7 | ID | 139.2 | 10 | 51.7 | 9.5 | 0 |
| 3 | Sex (by ID) | 140.4 | 9.9 | 52.9 | 9.9 | 0 |

**Table 3.** The diet composition (%) of male and female northern gannets in 2017 and 2018 as predicted by Bayesian mixed effects modelling, reported in table 2.

| species name | common name | 2017 | | 2018 | |
|---|---|---|---|---|---|
| | | female (%) | male (%) | female (%) | male (%) |
| *Ammodytes* spp. | sandeels | 13.3 | 13 | 4.5 | 4.5 |
| *Callionymus* spp. | dragonet | 4.4 | 5.5 | 5.8 | 7.7 |
| *Chelidonichthys cuculus* | red gurnard | 3.8 | 4.9 | 2.2 | 3 |
| *Clupea harengus* | Atlantic herring | 6 | 6.9 | 2.8 | 3.4 |
| *Merlangius merlangus* | whiting | 6.4 | 8.3 | 1.6 | 2.2 |
| *Merluccius merluccius* | hake | 6 | 6.9 | 4.2 | 4.6 |
| *Pleuronectes platessa* | plaice | 2.5 | 3 | 2.4 | 3.3 |
| *Scomber scombrus* | mackerel | 37.3 | 33.9 | 68.7 | 64.4 |
| *Sprattus sprattus* | sprat | 15 | 12.1 | 5.8 | 4.8 |
| *Trisopterus esmarkii* | Norway pout | 5.1 | 5.6 | 2 | 2.2 |

between the sexes coupled with resource partitioning to avoid intraspecific competition. SIA indicated sex-specific diets, but there was no difference in energy intake between the sexes, despite the difference in mass. The cost of individual prey capture attempts associated with differing diets was low compared with total energetic expenditure, and despite females diving more than males and being heavier, there was no difference in auxilliary energetic expenditure per day between the sexes.

The methodology presented here represents a simple and easily accessible way of calculating the energetic cost of specific behaviours from accelerometry data where allometric estimates of BMR and FMR are available. DBA is an established proxy measure of energy expenditure [83], though difficulties remain in converting DBA to a true measure of energy expenditure [26]. Studies comparing DBA with energy expenditure must ensure that summed values of energy expenditure must not simply be regressed against summed values of DBA through time, a problem known as the time trap [84,85]. In this study, we accounted for time by considering complete 24 h periods, allowing for meaningful estimates of energy expenditure per unit of DBA and conversion to temporal periods based on this conversion rate and total DBA. Though we do not account for the error of environmental influences, we have assumed that this variance is equal between individuals.

The resulting energetic cost of prey capture events was low, even after including the cost of take-off from the sea surface following a dive, most likely due to the very low daily dive rate and short duration of this behaviour. For all individuals, prey capture attempts across the time tagged accounted for less

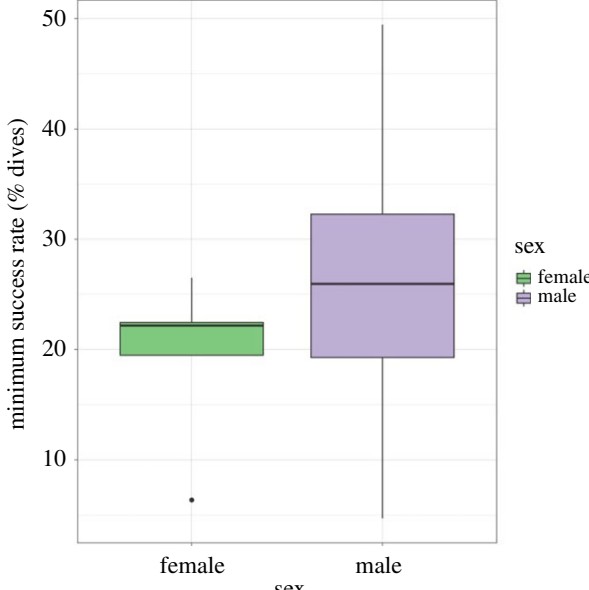

**Figure 3.** Minimum feeding success rates between the sexes to maintain body mass and provision a chick. Males were predicted to require a higher feeding success rate due to the lower numbers of dives undertaken. The middle horizontal line of the boxplot represents the median of the data range, boxes represent the 25th and 75th percentile with lines showing the remaining range of data (with outliers shown as dots).

than 4% of total energy expenditure. This suggests that the cost of diving probably does not limit the number of prey capture attempts in gannets from Great Saltee during our study period. This further suggests that gannets are not currently foraging at the limit of their energy demands. By contrast, little auks feeding on copepods were found to be required to feed upon six copepods a second to meet energy requirements [86].

Despite females undertaking an average of eight more dives per day, the low cost of prey capture attempts contributed to no differences in daily energetic expenditure between males and females. Year and dive type (plunge versus pursuit dive) had the largest effect on energetic cost of diving, reflecting yearly differences in diet noted in SIA analysis, that are probably related to the proportion of different dive types.

The cost of individual prey capture attempts may be slightly greater in females, as they spend more time underwater. However, energy expenditure can be affected by the medium an animal moves through [87], and this then may affect the sexes unevenly, though this is unlikely given the proportionally low energetic costs of diving. The increased cost of underwater pursuit following a 'failed' plunge dive suggests a cost-benefit trade-off, and Machovsky-Capuska *et al.* [88] noted higher feeding success in pursuit dives in Australasian gannets, *Morus serrator*, that would support this hypothesis. Alternatively, females may have to dive more as they are not as initially successful in the plunge dives; though our methodology only allows for minimum success rate to be calculated and this remains unknown. Intraspecific competition is expected to be higher with increasing proximity to a breeding colony [54,89] and this competition may drive sexually divergent foraging behaviour in gannets. Several studies report that male gannets forage closer to breeding colonies while females travel further [44,46]. This may be due to male gannets outcompeting females, forcing them to travel further and undertake different dive behaviour as they are forced to forage in different habitat to males [47,48] and this may also be a contributing factor to the different dive costs between the sexes reported in this study.

Gannets forage on a wide variety of prey [90], and SIA models indicated divergent diets between males and females, consistent with previous studies [44,48]. Prey proportions from our SIA models were similar to those previously reported by Lewis *et al.* [72] at the same site, and we found females took proportionately more mackerel and less whiting, Norway pout, and herring compared with males. After applying the average calorific content of prey species to sex-specific diets, energetic gain per dive did not differ between sexes. However, females make a greater contribution to chick provisioning [50], which may require a proportionate increase in targeting of smaller sized prey for chick consumption. While this has been observed in other seabird species [91], there is little evidence

**Table 4.** Summary of results from tracked northern gannets between 2017 and 2018. Individuals were tracked using a combination of accelerometry, time depth recorders and GPS. Energy expenditure is calculated from the approach outlined in figure 2 and chick demands are accounted for by the amount of energy required by a four-week-old chick. Modelled average kJ per successful dive includes results from a Bayesian mixed model from isotope analysis and is produced as a figure for each sex per year. The kJ value of a successful dive, the number of dives undertaken and the overall energetic demands are then used to consider how many dives must be successful for a northern gannet to survive and raise a chick.

| bird ID | sex | year of study | tracking duration (days) | number of dives | dives per day | total energy expenditure during tracking (kJ) | total energy expenditure during tracking plus chick demands (kJ) | energy expenditure per day with chick demands (kJ) | modelled average kJ per successful dive | number of successful dives to meet energy demands | per cent of recorded dives needed to be successful |
|---|---|---|---|---|---|---|---|---|---|---|---|
| D01 | male | 2017 | 4.9 | 113 | 23.06 | 17 024 | 19 763 | 4033 | 1005.04 | 19.66 | 17.4 |
| D02 | male | 2017 | 2.86 | 36 | 12.56 | 8940 | 10 538 | 3684 | 1005.04 | 10.48 | 29.13 |
| D03 | unknown | 2017 | 5.08 | 189 | 37.15 | 15 840 | NA | NA | NA | NA | NA |
| D04 | female | 2017 | 0.97 | 65 | 66.89 | 3341 | 4154 | 4282 | 1005.96 | 4.13 | 6.35 |
| D05 | female | 2017 | 1.83 | 39 | 21.31 | 6109 | 7643 | 4176 | 1005.96 | 7.6 | 19.48 |
| D12 | female | 2017 | 4.68 | 90 | 19.19 | 16 147 | 20 070 | 4288 | 1005.96 | 19.95 | 22.17 |
| D13 | male | 2017 | 4.72 | 87 | 18.44 | 14 761 | 17 399 | 3686 | 1005.04 | 17.31 | 19.9 |
| D16 | male | 2017 | 1.99 | 18 | 9.06 | 6063 | 7175 | 3605 | 1005.04 | 7.14 | 39.66 |
| D25 | female | 2018 | 3.04 | 37 | 12.17 | 10 436 | 12 984 | 4271 | 1563.34 | 8.31 | 22.45 |
| D26 | male | 2018 | 2.92 | 36 | 12.31 | 11 096 | 12 728 | 4359 | 1552.61 | 8.2 | 22.77 |
| D28 | male | 2018 | 4.61 | 230 | 49.84 | 14 155 | 16 731 | 3629 | 1552.61 | 10.78 | 4.68 |
| D41 | male | 2018 | 4.83 | 39 | 8.07 | 15 351 | 18 051 | 3737 | 1552.61 | 11.63 | 29.81 |
| D52 | female | 2018 | 4.32 | 42 | 9.72 | 13 786 | 17 408 | 4029 | 1563.34 | 11.14 | 26.51 |
| D53 | male | 2018 | 5.05 | 25 | 4.95 | 16 375 | 19 198 | 3801 | 1552.61 | 12.37 | 49.46 |

to suggest this is the case in northern gannets. Our results support that divergent diet is not the result of differing energetic cost of prey capture, or energy content of prey. Instead, sex differences in diet may be a result of intersexual competition, as previously demonstrated in this population of gannets [48].

Female gannets dived more frequently than males which may be reflective of differing provisioning roles [92,93], with female gannets estimated to have a 9.6% higher TED than male gannets, largely because of their greater contribution to chick feeding [50]. After accounting for the increased energetic demands in females, the energetic cost of foraging, the mean calorific content of prey in sex-specific diets and the number of dives performed, males were predicted to have a higher minimum feeding success rate than females (19% of dives in females and 26% of dives in males). These estimates of feeding success are lower than estimates of approximately 50–66% for Australasian gannets based on identifying prey captures from bird-borne cameras [94,95]. Our estimates reflect *minimum* success rates required to meet energy demands, and the discrepancy suggests that gannets may routinely catch more food than required to meet minimum energy demands. Any energy surplus may then be invested in chick provisioning, or be expended engaging in energetically demanding activities around and within the colony such as preening and aggression [96]. Energy surplus may then also be turned into body mass [97] or other costly procedures such as moulting and growth of new feathers.

Energy acquisition and allocation provide a useful framework to study ecological questions, including management and evolution [98]. This study highlights how DBA can be used to estimate the energetic costs of discrete short-lived behaviours, providing insights into the foraging ecology of free-ranging animals. While gannets are sexually monomorphic, they show divergent foraging behaviour and diet, which our results suggest are not the result of differing cost of foraging or energy content of prey. Instead, such sexually divergent foraging strategies in monomorphic species are thought to be driven by intersexual competition [48] or differing energy demands such as unequal parenting roles between the sexes [20]. Female gannets meet this additional need through increasing their dive rate, a strategy that has no appreciable additional cost given the small overall cost of individual dives and may be an adapted strategy to account for competitive exclusion. Over the course of a breeding season, this extra energetic expenditure equates to approximately 1567 kJ, less than the energy provided by one mackerel. However, after accounting for the cost of dives, the energetic content of prey, and the number of dives performed, females appear to have lower overall success rates to meet energetic requirements, suggesting some subtle difference in foraging behaviour that may represent competitive exclusion between the sexes [44,48] or another mechanism that remains unknown.

Our methodology and results have highlighted that in northern gannets, a sexually monomorphic species, the sexes show differences in foraging behaviour primarily related to dive rate and feeding success rather than the energetic cost of foraging. Evaluating sex differences in foraging behaviour from an energetic perspective may provide a clearer picture for understanding sexually divergent foraging strategies in both sexually monomorphic and dimorphic species. Future research should consider an energetics approach in exploring the fine-scale behavioural differences between sexes. It would be interesting to see this study replicated using dimorphic species, where differences between the sexes are more clearly pronounced, to see if sex differences in foraging behaviour may change with corresponding differences in energetic expenditure beyond those due to mass alone [99]. A further opportunity of study would be to consider how the sexes may differ in their energy expenditure with changing resources [100].

Ethics. All research was approved by the University College Cork Animal Ethics Committee and was conducted under licence from the Health Products Regulatory Authority, the National Parks and Wildlife Service and the British Trust for Ornithology.

Data accessibility. All data collected as part of this study are available from the Dryad Digital Repository: https://doi.org/10.5061/dryad.zs7h44j88 [51].

The data are provided in the electronic supplementary material [101].

Authors' contributions. A.B.: conceptualization, formal analysis, funding acquisition, investigation, methodology, writing—original draft and writing—review and editing; J.G.: formal analysis, methodology and writing—review and editing; J.L.Q.: funding acquisition, methodology, supervision and writing—review and editing; J.A.G.: investigation, methodology, validation and writing—review and editing; M.J.: conceptualization, data curation, funding acquisition, investigation, writing—original draft and writing—review and editing.

All authors gave final approval for publication and agreed to be held accountable for the work performed therein.

Competing interests. We declare we have no competing interests.

Funding. A.B. was funded by the Irish Research Council Postgraduate Scholarship (Project ID: GOIPG/2016/503).

Acknowledgements. We are grateful to the Neale family for access to Great Saltee where this work was undertaken. All research was approved by the University College Cork Animal Ethics Committee and was conducted under licence from the Health Products Regulatory Authority, the National Parks and Wildlife Service and the British Trust for Ornithology.

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
