## [Peer Review File · Royal Society Open Science]

Review History

RSOS-210520.R0 (Original submission)

Review form: Reviewer 1

Is the manuscript scientifically sound in its present form?

Yes

Are the interpretations and conclusions justified by the results?

Yes

Is the language acceptable?

Yes

Do you have any ethical concerns with this paper?

No

Have you any concerns about statistical analyses in this paper?

No

Recommendation?

Accept with minor revision (please list in comments)

Comments to the Author(s)

I really enjoyed reading this manuscript! Well done. The introduction was great and flowed nicely, providing a background into sex-specific differences in foraging behaviour, its potential drivers (energy expenditure and diet), the methods used to investigate this and your study system. This set the manuscript up well.

Are you able to provide any further information regarding the “marginal” sex differences in weight observed in gannets, that you mention within your introduction? Are the other sex differences in gannets that you mention (foraging behaviour and diet) considered to be ubiquitous, or are they only true of particular populations from/foraging in particular locations?

In your methods section, your field methods are good and descriptive but could include a bit more detail with regards to the potential for loggers to have impacted the gannets' behaviour/demographic parameters. Some would argue that total deployment weights as well as the weights of the birds (upon deployment and retrieval possibly) should be included. You also don't currently provide any methodology behind logger retrievals, including how many birds/loggers were recaptured and when this occurred (i.e., how long the deployment length was). I know that this is mentioned later within your results, but wonder whether it should be considered as more of a methodological point. I also wonder whether you should move your bloods methods to the Data Collection section, as I was surprised to read this section of text later on within your methods instead. Otherwise, perhaps you could rename your section heading methods so that they read “Biologging Data Collection” instead, or something similar.

I feel that Figure 1 is a conceptual diagram of your study methods, as opposed to the actual study and your specific hypotheses?

Please can you clarify the methods behind “confirming” TDR dives? Was this via visual inspection, as suggested in your Results?

I wonder whether it might be helpful to rearrange the order of your “Energetics from Accelerometry” section so that you first state what you are aiming to do with these methods, and then outline the steps that you took to achieve this goal.

I wonder whether the second paragraph of your “Statistical Analysis” section should feature earlier on as I'm not sure that it is really describing statistical analyses particularly. Perhaps this is also true of some of the following paragraph, i.e., the fish allometry etc.

When discussing sex differences in diving behaviour within your Results, perhaps considering including the percentage differences between some of the male and female metrics within the results would be helpful, rather than just the means of each sex.

I think that your table and figure headings could be more descriptive so that they are able to be easily interpreted as stand-alone items, without the remainder of the text being read. For example, you could include the species and colony that you are investigating.

I'm not sure whether I agree that there are not previous instances of the energetic cost of individual prey capture attempts being estimated in seabirds. Haven't seabird-mounted cameras

paired with accelerometry been used to do this? Or devices that record beak opening events/changes in oesophageal temperature? Maybe I'm wrong, but perhaps these are methods that could also be mentioned in your Introduction if trying to estimate the energetic cost of prey capture attempts is a key goal of this manuscript.

I think that your Discussion could generally do with another check through to ensure that the readability is as good as elsewhere in your manuscript and that it flows and covers all of the aspects that you want it to, in a way that flows and makes sense. For example, I'm not totally sure what the goal of the large second paragraph is at the moment as you discuss a number of different results in turn throughout. Additionally, I think that L413-20 in particular could be streamlined a little to increase their readability. I know what you're trying to say, but I think that they could benefit from a little more editing, including the mention of it being the sexes that have divergent diets within the final sentence of this paragraph. I've recommended some grammatical changes to L421-7 too, but also wonder whether you could tie this back to the results of this manuscript a bit more. The same is also true of the following paragraph (L428-34) and elsewhere within your Discussion.

Some of your in-text citations seem to be in a strange format and should be double checked throughout.

I've provided a marked-up document of the pdf (see Appendix A) with a few more small comments here and there, but otherwise, good job!

Decision letter (RSOS-210520.R0)

Dear Dr Bennison

The Editors assigned to your paper RSOS-210520 "A bioenergetics approach to understanding sex differences in the foraging behaviour of a sexually monomorphic species" have now received comments from reviewers and would like you to revise the paper in accordance with the reviewer comments and any comments from the Editors. Please note this decision does not guarantee eventual acceptance.

Please submit your revised manuscript and required files (see below) no later than 21 days from today's (ie 02-Sep-2021) date. Note: the ScholarOne system will 'lock' if submission of the revision is attempted 21 or more days after the deadline. If you do not think you will be able to meet this deadline please contact the editorial office immediately.

on behalf of Dr Agustina Gómez-Laich (Associate Editor) and Kevin Padian (Subject Editor)
openscience@royalsociety.org

Associate Editor Comments to Author (Dr Agustina Gómez-Laich):

Associate Editor: 1

Comments to the Author:

Dear authors,

In this study, the authors used a bioenergetic approach to examine intersexual differences in the foraging behavior of a sexually monomorphic seabird; the Northern Gannet. To do this, they instrumented female and male breeding gannets with GPSs, accelerometers, and TDRs. Energy expenditure was estimated using dynamic body acceleration and afterward converted to kilojoules. Additionally, Stable Isotope Analyses was employed to study intersexual differences in diet and to estimate each sex's energy acquisition. The study's main finding is that sex differences in foraging behavior are mainly associated with dive rate and success. This study provides an important contribution to science, but the methods and discussion sections need to be improved. I feel authors should address some general and specific issues (see below) before the manuscript can be published.

General comments

- 1) Please present in a more explicit way the hypothesis you are testing. This would help to structure the manuscript. Additionally, it would greatly improve the manuscript if authors refer to the main questions and hypotheses/predictions along with the methodology, results, and discussion sections. For example, several sections of the discussion refer to the methodology employed to estimate energy expenditure. Even though this is an important aspect of the Ms, it is not one of the main objectives.
- 2) Several sections of the methodology need to be better explained (see below).
- 3) Results could be improved by incorporating more tables for example as Supplementary material.
- 4) The discussion could be greatly improved if authors focused more on debating their findings and how they contribute to the main hypothesis. Many paragraphs repeat information that was already mentioned in the results.

Specific comments

Keywords: accelerometry instead of accelerometry

Keywords: Isotope or isotope?

Introduction

Lines 77-79. It would be interesting to explain how foraging at different locations may benefit females to restore their body condition after egg production.

Lines 79-81. Can you give an example of how sex-specific foraging strategies in sexually monomorphic species may be driven by intraspecific competition causing one sex to be displaced spatially or to forage in different niches?

Lines 76-86. In this second paragraph, the first and fourth sentences present very similar ideas.

Line 102. Please give examples of other direct methods.

Line 103. Can you briefly explain what each isotopic ratio allows us to know about prey consumption?

Line 107. Please add what this marginally heavier means.

Line 114-115. Than forage fish female specialist sounds awkward.

Line 119-125. I suggest rephrasing this last paragraph. First, mention the objectives and then the methodology you will use to reach them. Additionally, It would be interesting to be more explicit with the hypothesis you are testing (see general comments). It would also be interesting to present some predictions.

Methodology

Line 140. Figure 1 does not show the conceptual diagram of the study. This figure shows a diagram of a part of the methodology.

Line 140. How many males, and how many females were instrumented? Did you instrument pairs? How much time were the devices left on the birds? This information is mentioned in the results instead of being in the methodology. Did birds that were instrumented continued breeding normally?

Line 141. This means that in some nests chicks were 21 days old and in other chicks were more than a month? Is the foraging behavior similar during both stages of the breeding season (3 vs 5 weeks old chicks)? Please incorporate information regarding this particular topic.

Please add the dimensions of each one of the loggers you deployed on the birds.

Line 152. How much blood was taken? From where was the blood sample taken?

Figure 1. I suggest including which were the defined behaviors.

Line 159-160. "further methods that develop the findings" sounds awkward. Please rephrase.

Line 162. Please explain more in detail how you analyze the acceleration data. To obtain a time-activity budget and estimate the energy expenditure during a foraging trip, not only the dives should be identified from the acc data. How did you recognize when birds were flying and floating? Equations to estimate energy expenditure from VeDBA may be activity-specific. For

example, the equation used to derive energy expenditure for flying and diving may be different. Was this taken into account? Please give more details about this particular subject. Additionally, please give more details about how the acceleration data was processed. Once you had the acceleration data, you calculated the average value for each of the axes using a running mean of 2 seconds? How did you calculate the pitch values from the acc data? What does the X, Z, and Y-axis mean? Please specify which is the heave, sway and surge.

Lines 172-175. It would be nice to see a figure of the bimodal distribution accelerometer-derived dives had as supplementary information.

Line 177. To do this first you had to obtain the time-activity budget of the birds, that is to say, how much time each bird expended on each activity. For this, the acceleration data such me labeled. How did you do this? Visually? Using some algorithm? Please give more details about this.

Line 179. Please explain how you calculated VeDBA. It is not clear if you calculated the VeDBA for each activity (flying, floating, diving) and once this had been done using a specific equation to convert this VeDBA value into kilojoules. Which allometric equations did you use? It would be worth incorporating this as supplementary information also.

Line 191. What does the individualized VeDBA to kJ equation mean?

Line 193. Couldn't you use the TDR information to determine when birds were on the surface?

Line 204. n=19 in 2017 and n=28 in 2018 should be placed after 47 birds.

Line 234. Here it says LMER but in the results it says GLM. Did you perform an LMER or a GLM? Please add which distribution was used and why.

Line 235. Please explain why the interaction between year and dive type was included in the model.

Line 239. In general, there seems to be some controversy about model averaging. If your top model has relatively good support (as compared to second-best models) some suggested it may be better to refrain from model averaging. Why did you choose to do model averaging? Please explain how you obtained the average. It would be interesting to incorporate a table with the best models, their AIC, deltaic, and weight as supplementary material.

Line 246. These sentences are not clear. It is not clear if you used trips as energetic units or you also considered some periods at the colony.

Line 247. So you calculated a foraging trip energy expenditure and a 24 hour period energy expenditure? Please clarify this aspect. Comparisons between females and males were performed for both time periods (foraging trip and 24-hour period)?

Line 254. Did you mean the total amount of food they needed to eat to get the energy they expended? Cant this be achieved by eating more than one combination of prey proportions?

Line 267-268. Please rephrase this sentence. It is not clear.

Results.

Line 282. Does this mean that from one individual you couldn't determine the sex from the blood sample?

Line 290. Please explain better how this average LMER was obtained. It would be nice to see in a table all the models that showed delta AIC values higher than 6.

Line 316. Why here you present a Chi-square and in Line 313 an F? How did you get these statistics? This is not mentioned in the methodology.

Figure 3. Please explain what each part of the boxplot means.

Discussion

Line 385-387. This sentence is not clear. I would rather say that you focused on behaviors that imply movement. Certain behaviors do not imply movement and for that behaviors, VEDBA would not be useful.

Line 399-401. Energy expenditure can be affected by the medium in which an animal moves especially if the movement in different media involves different muscle groups.

Line 449. It would be nice to see a table showing how much energy each bird expended in the different behaviors that comprise the foraging trip. In this table, the time engaged in each behavior could also be included.

Line 470-471. Can you please more information and specific examples about how the bioenergetic approach presented in this study could contribute to future studies?

Associate Editor: 2

Comments to the Author:
(There are no comments.)

Reviewer comments to Author:

Reviewer: 1

Comments to the Author(s)

I really enjoyed reading this manuscript! Well done. The introduction was great and flowed nicely, providing a background into sex-specific differences in foraging behaviour, its potential drivers (energy expenditure and diet), the methods used to investigate this and your study system. This set the manuscript up well.

Are you able to provide any further information regarding the “marginal” sex differences in weight observed in gannets, that you mention within your introduction? Are the other sex differences in gannets that you mention (foraging behaviour and diet) considered to ubiquitous, or are they only true of particular populations from/foraging in particular locations?

In your methods section, your field methods are good and descriptive but could include a bit more detail with regards to the potential for loggers to have impacted the gannets' behaviour/demographic parameters. Some would argue that total deployment weights as well as the weights of the birds (upon deployment and retrieval possibly) should be included. You also don't currently provide any methodology behind logger retrievals, including how many birds/loggers were recaptured and when this occurred (i.e., how long the deployment length was). I know that this is mentioned later within your results, but wonder whether it should be considered as more of a methodological point. I also wonder whether you should move your bloods methods to the Data Collection section, as I was surprised to read this section of text later

on within your methods instead. Otherwise, perhaps you could rename your section heading methods so that they read “Biologging Data Collection” instead, or something similar.

I feel that Figure 1 is a conceptual diagram of your study methods, as opposed to the actual study and your specific hypotheses?

Please can you clarify the methods behind “confirming” TDR dives? Was this via visual inspection, as suggested in your Results?

I wonder whether it might be helpful to rearrange the order of your “Energetics from Accelerometry” section so that you first state what you are aiming to do with these methods, and then outline the steps that you took to achieve this goal.

I wonder whether the second paragraph of your “Statistical Analysis” section should feature earlier on as I’m not sure that it is really describing statistical analyses particularly. Perhaps this is also true of some of the following paragraph, i.e., the fish allometry etc.

When discussing sex differences in diving behaviour within your Results, perhaps considering including the percentage differences between some of the male and female metrics within the results would be helpful, rather than just the means of each sex.

I think that your table and figure headings could be more descriptive so that they are able to be easily interpreted as stand-alone items, without the remainder of the text being read. For example, you could include the species and colony that you are investigating.

I’m not sure whether I agree that there are not previous instances of the energetic cost of individual prey capture attempts being estimated in seabirds. Haven’t seabird-mounted cameras paired with accelerometry been used to do this? Or devices that record beak opening events/changes in oesophageal temperature? Maybe I’m wrong, but perhaps these are methods that could also be mentioned in your Introduction if trying to estimate the energetic cost of prey capture attempts is a key goal of this manuscript.

I think that your Discussion could generally do with another check through to ensure that the readability is as good as elsewhere in your manuscript and that it flows and covers all of the aspects that you want it to, in a way that flows and makes sense. For example, I’m not totally sure what the goal of the large second paragraph is at the moment as you discuss a number of different results in turn throughout. Additionally, I think that L413-20 in particular could be streamlined a little to increase their readability. I know what you’re trying to say, but I think that they could benefit from a little more editing, including the mention of it being the sexes that have divergent diets within the final sentence of this paragraph. I’ve recommended some grammatical changes to L421-7 too, but also wonder whether you could tie this back to the results of this manuscript a bit more. The same is also true of the following paragraph (L428-34) and elsewhere within your Discussion.

Some of your in-text citations seem to be in a strange format and should be double checked throughout.

I’ve provided a marked-up document of the pdf with a few more small comments here and there, but otherwise, good job!

===PREPARING YOUR MANUSCRIPT===

===PREPARING YOUR REVISION IN SCHOLARONE===

Author's Response to Decision Letter for (RSOS-210520.R0)

See Appendix B.

Decision letter (RSOS-210520.R1)

Dear Dr Bennison

On behalf of the Editors, we are pleased to inform you that your Manuscript RSOS-210520.R1 "A bioenergetics approach to understanding sex differences in the foraging behaviour of a sexually monomorphic species" has been accepted for publication in Royal Society Open Science subject to minor revision in accordance with the referees' reports. Please find the referees' comments along with any feedback from the Editors below my signature.

Please submit your revised manuscript and required files (see below) no later than 7 days from today's (ie 14-Dec-2021) date. Note: the ScholarOne system will 'lock' if submission of the revision is attempted 7 or more days after the deadline. If you do not think you will be able to meet this deadline please contact the editorial office immediately.

on behalf of Dr Agustina Gómez-Laich (Associate Editor) and Kevin Padian (Subject Editor)
openscience@royalsociety.org

Associate Editor Comments to Author (Dr Agustina Gómez-Laich):

Specific comments to authors.

Introduction.

Line 88. Please incorporate the Brown booby specific species name.

Lines 137-144. I realize in the previous version my suggestion was to first state the objectives and afterwards the technology employed. In the present version I suggest first mentioning the main objective and methodology employed and afterwards mention the specific objectives. For example:

In the present study, we used GPS, accelerometry, and SIA data to gain a better understanding of how gannets engage in foraging and how different demands upon the sexes may affect foraging strategies. Specifically, we explore sex differences in foraging of gannets in terms of diet, dive types, frequency of prey capture attempts, and the energetic cost of prey capture attempts. Additionally, we quantify the energetic requirements of each sex, taking into account energy expended during foraging and, using data from published studies, energetic demands of feeding offspring. Finally, we consider minimum dive success rates necessary for male and female gannets to meet their energy demands.

Lines 151-157. Hypothesis. The first one is fine however, the second and the third one are predictions not hypotheses. Please rephrase them.

Methods

Line 174. Please revise the numbers, here the total number of instrumented birds is 14 and below is 13. Please state how many females and males were equipped each year.

Line 175. Please change for 52° 7' 37.92" N, 6° 35' 45.6" W

Line 181. Which was the depth threshold? 0.5 or 1 m? Or some devices were programmed with a 0.5 threshold and some with a 1 m threshold? Please clarify this aspect.

Line 189. Is a period missing after (52)? The following that starts with "Previous" sounds a bit awkward, please rephrase it.

Line 189. the "s" in gannets looks like a subscript letter.

Line 192. This is the first time a table of the Supplementary information is mentioned so I suggest considering this table as table S1 instead of table S5. Please check that all Supp. table numbers are correctly mentioned in the main document after they are renumbered.

Line 268. For example, this would be table S2 now.

Line 293-297. This sentence is too long and not clear. Please rephrase it.

Line 336. Please check Supplementary information Table numbers.

Line 368. It is not clear to me how you test for differences in diving rate between sexes using linear regression. Please clarify this aspect.

Results.

Line 421. Please mention in the methods how you tested for differences in body mass between sexes.

Line 426. Why didn't you test for differences in dive duration between sexes?

Line 428. Why didn't you test for differences in dive + take off costs between sexes?

Line 461. In the methods, you mention that differences in the diving rate between sexes were tested by means of linear regression and here a GLM is mentioned. Please clarify this aspect.

Line 501. You can say KIV instead of "average energy intake (KIV)" since you have already defined what KIV stands for.

Discussion

Line 586. Please eliminate "do" from "Females may have to do dive more".

Line 596. "gannets" can be eliminated here since it is clear you are talking about gannets.

Reviewer comments to Author:

===PREPARING YOUR MANUSCRIPT===

one version should clearly identify all the changes that have been made (for instance, in coloured highlight, in bold text, or tracked changes);

Please ensure that you include an acknowledgements' section before your reference list/bibliography. This should acknowledge anyone who assisted with your work, but does not

qualify as an author per the guidelines at <https://royalsociety.org/journals/ethics-policies/openness/>.

===PREPARING YOUR REVISION IN SCHOLARONE===

- Ensure that your data access statement meets the requirements at <https://royalsociety.org/journals/authors/author-guidelines/#data>. You should ensure that you cite the dataset in your reference list. If you have deposited data etc in the Dryad repository, please only include the 'For publication' link at this stage. You should remove the 'For review' link.
- If you are requesting an article processing charge waiver, you must select the relevant waiver option (if requesting a discretionary waiver, the form should have been uploaded, see 'File upload' above).
- If you have uploaded any electronic supplementary (ESM) files, please ensure you follow the guidance at <https://royalsociety.org/journals/authors/author-guidelines/#supplementary-material> to include a suitable title and informative caption. An example of appropriate titling and captioning may be found at https://figshare.com/articles/Table_S2_from_Is_there_a_trade-off_between_peak_performance_and_performance_breadth_across_temperatures_for_aerobic_scope_in_teleost_fishes_/3843624.

Author's Response to Decision Letter for (RSOS-210520.R1)

See Appendix C.

Decision letter (RSOS-210520.R2)

Dear Dr Bennison,

I am pleased to inform you that your manuscript entitled "A bioenergetics approach to understanding sex differences in the foraging behaviour of a sexually monomorphic species" is now accepted for publication in Royal Society Open Science.

on behalf of Dr Agustina Gómez-Laich (Associate Editor) and Kevin Padian (Subject Editor)
openscience@royalsociety.org

Appendix A**ROYAL SOCIETY
OPEN SCIENCE****A bioenergetics approach to understanding sex differences
in the foraging behaviour of a sexually monomorphic
species**

Journal:	Royal Society Open Science
Manuscript ID	RSOS-210520
Article Type:	Research
Date Submitted by the Author:	13-Apr-2021
Complete List of Authors:	Bennison, Ashley; Galway-Mayo Institute of Technology, Earth & Environmental Sciences; University College Cork, School of Biological, Earth & Environmental Sciences; University College Cork, MaREI Giménez, Joan; University College Cork, MaREI Quinn, John; University College Cork, School of Biological, Earth, & Environmental Sciences Jessopp, Mark; University College Cork National University of Ireland, MaREI Centre, Environmental Research Institute; University College Cork National University of Ireland, School of Biological, Earth & Environmental Sciences
Subject:	ecology < BIOLOGY, bioenergetics < CROSS-DISCIPLINARY SCIENCES
Keywords:	bioenergetics, sex differences, seabirds, movement ecology
Subject Category:	Organismal and Evolutionary Biology

Author-supplied statements

Relevant information will appear here if provided.

Ethics

Does your article include research that required ethical approval or permits?:

Yes

Statement (if applicable):

All research was approved by the University College Cork Animal Ethics Committee and was conducted under licence from the Health Products Regulatory Authority, the National Parks and Wildlife Service, and the British Trust for Ornithology.

Data

It is a condition of publication that data, code and materials supporting your paper are made publicly available. Does your paper present new data?:

Yes

Statement (if applicable):

Data and scripts have been uploaded to Dryad and are available at:

<https://doi.org/10.5061/dryad.zs7h44j88>

A link for access to review the dataset is provided below:

<https://datadryad.org/stash/share/PlzjD8vxRQvP7RqRYlvZfd6j88Mc4s6bGnOEJfnon6Y>

Conflict of interest

I/We declare we have no competing interests

Statement (if applicable):

CUST_STATE_CONFLICT :No data available.

Authors' contributions

This paper has multiple authors and our individual contributions were as below

Statement (if applicable):

AB, MJ, and JQ conceptualised the project. AB and MJ undertook fieldwork to collect data. JG undertook preparation of isotopic samples and isotope modelling. AB undertook the remaining analysis. All authors contributed actively the writing of the manuscript and approve the final edit.

A bioenergetics approach to understanding sex differences in the foraging behaviour of a sexually monomorphic species

Ashley Bennison^{1,2,3}, Joan Giménez², John L. Quinn³, Mark Jessopp^{2,3}

1. Marine and Freshwater Research Centre, Galway-Mayo Institute of Technology, Galway, Ireland
2. Centre for Marine Renewable Energy, College of Science, Engineering and Food Science, University College Cork, Ireland
3. School of Biological, Earth & Environmental Sciences, College of Science, Engineering and Food Science, University College Cork, Ireland

Corresponding Author: Ashley.bennison@gmit.ie

Data Accessibility

Data will be made available for access on the Dryad Open Access Repository once published.

Acknowledgements

We are grateful to the Neale family for access to Great Saltee where this work was undertaken. AB was funded by the Irish Research Council Postgraduate Scholarship (Project ID: GOIPG/2016/503). All research was approved by the University College Cork Animal Ethics Committee and was conducted under licence from the Health Products Regulatory Authority, the National Parks and Wildlife Service, and the British Trust for Ornithology.

**Author Contributions**

AB, MJ, and JQ conceptualised the project. AB and MJ undertook fieldwork to collect data. JG
undertook preparation of isotopic samples and isotope modelling. AB undertook the remaining
analysis. All authors contributed actively the writing of the manuscript and approve the final
edit.

**Keywords**

Northern gannet, Isotope ecology, movement ecology, bioenergetics, acclerometry

Abstract

[revised manuscript text omitted]

**Methods**

***Data Collection***

A conceptual diagram of our study is presented in Figure 1. Breeding adult gannets (n=8 in 2017,
n=6 in 2018) attending 3-5 week-old chicks were tracked from Great Saltee, south-east Ireland
(52° 7' 37.92", -6° 35' 45.6"). Birds were caught using an 8-10m pole with a metal crook, weighed,
and equipped with a combination of dataloggers. GPS loggers (i-gotU GT-120, Mobile Action
Technology Inc., Taipei, Taiwan, 14g) recorded locations every 3 minutes; time depth recorders
(TDR, CEFAS G5, 2.5g) recorded depth at 4Hz after exceeding a 0.5 or 1m depth threshold; tri-
axial accelerometers (Gulf Coast Data Concepts X16-mini, 17g) recorded *g*-forces ($1g =$
9.807m/sec^2) at 50Hz. GPS and TDR loggers were attached ventrally to 2-3 central tail feathers
using strips of waterproof Tesa tape. Accelerometers were attached to 10-15 mantle feathers
between the wings. Three birds in 2017 and six birds in 2018 were equipped with GPS, TDR, and
accelerometers, while the remaining birds were equipped with only GPS and accelerometers.
Total instrument mass was <2% of body mass, and positioned to minimise impact on gannet
movement, both aerodynamic and hydrodynamic (48). A small volume of blood was sampled for
stable isotope analysis (see below) and 2-3 breast feathers were plucked for genetic sexing
following the method outlined by Griffiths, Double (49).

**Figure 1.** Schematic of methodology for data processing and the steps required to explore the
 sex differences in the foraging of northern gannets. The process starts at top with the red box
 labelled “raw accelerometry data” and ends with the green box “Sex differences in foraging

ecology.” Blue boxes represent the methodology for analysing data and orange boxes represent
further methods that develop the findings.

***Data processing and dive behaviour definition***

Behaviour classification from accelerometry data used a thresholding approach. Thresholds were
determined using protocols and guidance set out by Collins, Green (50) and Shepard, Wilson (51).
Diving events occurred when average acceleration (running average of 2 seconds) in the X-axis
was $<0g$ and standard deviation (SD) in the mean X-axis was $>1.4g$. The end of a dive was defined
by a 1-second lagged maximum of pitch change within a 60 second period from the start of a
dive. Take-off events were defined with a threshold where, following a dive, the SD of the Z-axis
was $>1.8g$ and the SD of the X-axis was $>1g$. Take-off events were considered to have ended and
returned to normal flight when the SD of the Z-axis resolved to $<1.4g$ and the SD of the X-axis
was $<1.4g$.  subset of birds (n=9) tagged with both TDRs and accelerometers were used to
validate accelerometer-derived dive events by visually comparing timestamps to TDR confirmed
dives. Accelerometer-derived dives had a bimodal distribution and were split into plunge dives
and pursuit dives based on a distinct break within the frequency distribution at 5 seconds; plunge
dives are dives followed by an almost immediate rise to the surface, whilst a pursuit dive is
characterised by sustained chase of prey underwater.

***Energetics from Accelerometry***

Dynamic body acceleration (DBA) is a relative metric that can be used as a proxy for energetic
expenditure from animal movement (52), and can be used to develop highly accurate activity
budgets (53). We used Vectorial DBA (VeDBA) to account for any variation in tag alignment (54).
We converted VeDBA into kilojoules (kJ) using published data and allometric equations (55). The
relationship between energy expenditure (kJ) and VeDBA is linear amongst a variety of animal
taxa, including mammals, reptiles, and birds (26, 33, 56), with slope k . Using the process outlined
in Figure 2, it is possible to produce estimates of kilojoules expended in movement for given
periods or behaviours. The process assumes that energy expended in movement is equal to an
animal's Field Metabolic Rate (FMR) minus Resting Metabolic Rate (RMR) for any given period.
Totalled 24-hour VeDBA is therefore equivalent to kJ from FMR-RMR, assuming where VeDBA =

0, kJ = 0. Simple algebra can then produce a formula for kJ of any VeDBA score over any time
 period. Here, we used FMR estimates for Northern gannets provided by the Shiny App by Dunn,
 White (57), corrected for individual bird weight and colony latitude (52°N), and RMR estimates
 provided by allometric equations from Schmidt-Nielsen and Knut (55) to produce individualised
 VeDBA to kJ equations. This method aims to produce whole sum approaches to energy
 expenditure; at present it is not possible to effectively account for periods of rest on water where
 sea swell may predict energy via acceleration and we assume that all acceleration is from animal
 movement.

For Kilojoules in seconds from VeDBA:

$$\sum_{24}^0 \text{VeDBA} = \text{FMR} - \text{RMR} \rightarrow \text{Kj} \cdot \text{n}^{-1} = k \left(\frac{\sum_n^0 \text{VeDBA}}{n} \right)$$

$$\rightarrow \text{Kj} \cdot \text{s}^{-1} = k \left(\frac{\sum_{86400}^0 \text{VeDBA}}{86400} \right)$$

Sources:
 1) FMR from Dunn et al 2018
 2) RMR from Schmidt Nielsen & Knutt 1984

**Figure 2.** Conceptual diagram demonstrating how to extract the kilojoule values of any given
 specific behaviour or time period within an accelerometry dataset for energy consumed only by
 movement.

***Isotopic Analysis for Diet Composition***

Blood samples were taken from 47 birds, including the accelerometer-equipped birds, (n=19 in
2017 and n=28 in 2018) and used to construct a population model of dietary intake from isotope
analysis. Blood samples were taken from the tarsus during tag deployment and centrifuged for
10 minutes to separate red blood cells (RBC) from plasma. While RBC therefore represent diet
prior to the deployment, preliminary sampling showed that isotopic signatures do not differ
significantly between blood samples collected on deployment and recovery of devices
approximately 1 week apart (unpublished data). Stable Isotope Analyses were performed at
Elemtex UK (Stable Isotope and & Elemental Analysis Expertise), using a Thermoquest EA1110
Elemental Analyser linked to a Sercon 2020 stable isotope ratio mass spectrometer running in
continuous flow mode. Accuracy and precision were monitored through laboratory internal
standards and an in-house comparison standard nested within samples.

Prey stable isotope values were obtained from a published dataset of Celtic Sea fish samples (58).
These authors conducted stable isotope analysis of samples without lipid extraction. Then, the
$\delta^{13}\text{C}$ data included in the published data set are not corrected for differences in lipid content, but
the % C and N data was used to make the required corrections following Logan, Jardine (59). As
recommended by Phillips, Inger (60), a reduced prey dataset was used and included only those
species previously recorded in more than 3% of the diet for Great Saltee gannets (61). These
species can be seen in Table S1 and 3.

Using Bayesian isotopic mixed models, it was possible to compare blood values to reference prey
values to reconstruct diet of gannets. The model was run on “long” settings (chains = 3, length =
300000, burn-in = 2000000, thinning = 100), using average diet-to-tissue discrimination factors
(2.25 ± 0.61 ‰ for $\delta^{15}\text{N}$ and 0.24 ± 0.79 ‰ for $\delta^{13}\text{C}$) from various studies of piscivorous birds (62-
65). Model convergence was assessed with the Gelman-Rubin diagnostic (66). Sex-based diet
estimates were obtained through Bayesian mixing models using the R package ‘MixSIAR’ (67).
We fit several models of diet with fixed and random effects as covariates, and evaluated the
relative support for each model using LOO (leave-one-out cross-validation) weights (68). Model

outputs were then used to construct prey proportions in the diet of males and females in 2017
and 2018.

***Statistical analysis***

To explore sex differences in the overall cost of prey capture attempts (dive and subsequent take-
off), a linear mixed effect regression (LMER) was used to test for sex differences in dive and take-
off characteristics. Factors included year, sex, weight, dive type, and the interaction between sex
and dive type to predict kilojoules expended. Individual was included as a random effect to
account for repeated measures of the same individual. To select the most parsimonious model,
the dredge function from the 'MuMin' package was used (69). Any models within 6 AIC values
were kept and model averaging undertaken (70).

We used the relationship between VeDBA and kJ shown in Fig. 2 to estimate total energetic
*expenditure* for an individual bird from the time it left the colony, to the point of recapture.
Gannet trips may range from one to several days and so this whole sum approach allowed our
predictions to account for a full range of behaviours, from in colony, transiting, and foraging. As
gannet foraging trips may last several days, they incur increasing energetic costs during a foraging
trip such as feeding chicks upon return, we have included this in the analysis by considering
energetic differences from a whole sum approach and use gannet trips as the energetic unit. We
also consider individual energy expenditure per 24-hour period. We then calculated energetic
*demands* by adding to this value the energetic demand of raising a four week old chick of 1397.14
249 kJ/day (Montevecchi, Ricklefs (71), with females contributing 60% of this cost due to unequal
chick feeding in gannets (71, 72). Though it would be most appropriate to have information on
feeding rates of chicks in this study, we do not have this information and instead consider the
overall energy requirements of chicks which act as a proxy to feeding rates. This produced a value
of Total Energetic Demand (TED) for each gannet for the time it was tracked. Using sex-based SIA
model outputs, we predicted the proportion of prey species in the diet of male and female
gannets.

We assumed the sizes of individual prey species were similar to those in Lewis, Sherratt (61), a
study from the same colony that did not identify any difference in the size of fish caught between

the sexes. The size and mass of the fish were then used to calculate the kJ value of each fish
species (using allometric equations referenced by Lewis, Sherratt (61) and assuming a 76.1%
assimilation efficiency following Cooper (73), See Table S1). For each sex-specific diet, the energy
content (kJ) of each fish was multiplied by the proportions of species in the diet and these
proportional values were summed to provide an average kJ intake value (KIV) for a successful
dive (a dive resulting in prey capture) for each individual gannet, assuming that successful prey
capture results in capture of one prey item. A Mann-Whitney-Wilcoxon test was used to test for
differences in KIV between sexes. For each gannet, TED was divided by KIV to determine how
many successful dives were required to maintain body condition, forage, and provision for a
chick. This number was then used as a proportion of dives recorded to derive realistic individual
minimum prey capture rates.

Results

Of the 14 gannets tracked, five were female, eight were male, and one was of unknown sex. The
individual of unknown sex was not included in analysis of sex differences. Male gannets were on
average lighter than females; male weight was $2.70\text{kg} \pm 0.19$ with females weighing $2.99\text{kg} \pm 0.15$
(Wilcoxon test: $W=35.5$, $r=0.88$, $p=0.025$).

Sex differences in dive behaviour

1046 visually validated dives and subsequent take-off events were detected. 24% of dives were
pursuit dives with females having a slight tendency towards increased pursuit dives compared to
males. Combined cost of a single prey capture attempt (dive + take-off) in females was 2.17
$\pm 0.73\text{kJ}$ while for males it was $1.97 \pm 0.92\text{kJ}$. An averaged LMER indicated a significant effect of
dive type and year on energy expenditure associated with dives while sex was retained as a non-
significant factor (Table 1). The estimates for cost of all prey capture attempts represent $< 4\%$ of
the daily total energy expenditure for each individual. Accounting for unequal provisioning of the
chick, and the cost of foraging, daily energetic demands were 9.6% higher for females than males
(female TED = $4601\text{kJ} \pm 121.60$; male TED = $4207\text{kJ} \pm 278.37$, Wilcoxon test: $W=34$, $p<0.05$, total
number of female days: 14.84, total number of male days: 31.88).

**Table 1.** Conditional model summary from the averaged mixed effect linear regression used to
 predict kilojoules (kJ) expended during a prey capture attempt. Input variables were year (2017
 and 2018), sex (male and female), dive type (pursuit or plunge), and weight. The interaction
 between sex and dive type was also included. Dive type (plunge) and sex (Female) were absorbed
 into the intercept.

Dive energetics model	Coefficient	Std. Error	Adjusted SE	Z value	P value
Intercept	-1976.374	794.301	795.059	2.486	0.01293
Type(Pursuit)	0.6008	0.0480	0.0481	12.480	<0.001
Year	1.0327	0.330	0.331	3.117	<0.01
Weight	0.1837	0.856	0.857	0.214	0.8302
Sex(Male)	-0.0895	0.411	0.411	0.217	0.8278

Females undertook significantly more dives per day than males (25.9 and 17.3 respectively, GLM
 $F_{13}=8.63$, $p<0.01$). However, because the cost of individual prey capture attempts is so low, a
 linear mixed effect regression predicting the energy expenditure (kJ) per day for each individual
 from sex and year, with ID as a random effect, found no significant effect of sex on daily energy
 expenditure (LMER $\text{Chi}^2_{38} = 0.0004$, $p = 0.98$)

*Isotopic analysis*

The isotope mixing model predicted that the most consumed prey species were Atlantic mackerel
 (*Scomber scombrus*) (27.83 % ± 4.34) and European sprat (*Sprattus sprattus*) (19.16% ± 2.06)
 followed by Lesser sandeel (*Ammodytes marinus*) (11.47 % ± 0.99) and Atlantic herring (*Clupea*
 *harengus*) (11.26 % ± 1.46). The remaining species included in the models were each predicted to

contribute less than 8% to the overall diet. Seven different models were tested (Table 2) and the
 best model included *Year* as a covariate (Model weight: 76.8 %, Model 4). The second-best model
 included *Sex* and *Year* as variables with a relative weight of 23.1%, and was used to predict sex-
 specific diets in each study year. There was no support for a model using individual ID only. Diet
 between the sexes was similar in both years (Table 3), though mackerel made a higher
 contribution to male diet (difference of 3.4% in 2017 and 4.3% in 2018). In 2018 the predominant
 species consumed was mackerel (68.7% and 64.4% of diet for males and females respectively).

**Table 2.** Bayesian mixed effect model outputs to determine predictors of diet. The best model
 lent support for a Year only model, however the second-best model was Sex +Year with a model
 weight of 23.1%. This model was used to predict diet of the sexes. Leave One Out cross validation
 Information Criteria (LOOic) were used to assessed model suitability.

Model	Variables	LOOic	Standard error LOOic	Delta LOOic	Standard error delta LOOic	weight
4	Year	87.5	11.8	0	NA	0.768
6	Sex + Year	89.9	11.6	2.4	3	0.231
5	Year (by ID)	106.8	8.6	19.3	6.4	0
2	Sex	109.7	10.9	22.2	6	0
1	Null	110.7	11	23.2	5.5	0
7	ID	139.2	10	51.7	9.5	0
3	Sex (by ID)	140.4	9.9	52.9	9.9	0

**Table 3.** The diet composition (%) of males and females in 2017 and 2018 as predicted by
 Bayesian mixed effects modelling as reported in Table 2.

Species Name	Common Name	2017		2018	
		Female (%)	Male (%)	Female (%)	Male (%)
Ammodytes spp.	Sandeels	13.3	13	4.5	4.5
Callionymus spp.	Dragonet	4.4	5.5	5.8	7.7
Chelidonichthys cuculus	Red Gurnard	3.8	4.9	2.2	3
Clupea harengus	Atlantic Herring	6	6.9	2.8	3.4
Merlangius merlangus	Whiting	6.4	8.3	1.6	2.2
Merluccius merluccius	Hake	6	6.9	4.2	4.6
Pleuronectes platessa	Plaice	2.5	3	2.4	3.3
Scomber scombrus	Mackerel	37.3	33.9	68.7	64.4
Sprattus sprattus	Sprat	15	12.1	5.8	4.8
Trisopterus esmarkii	Norway Pout	5.1	5.6	2	2.2

Applying average energy content of prey in proportion to its occurrence in the diet, a successful
 dive was estimated to have an average energy intake (KIV) of 1006 kJ for females, and 1005 kJ
 for males in 2017. In 2018, this figure rose with changing diet to 1563 kJ for females and 1553 kJ
 for males.

Based on the number of dives performed and average energy content of prey in proportion to
 their occurrence in sex-specific diets, female minimum feeding success rate was calculated as
 21.21% \pm 8.42, whilst the male rate was 29.22% \pm 15.10 (Fig. 4.3). A summary of all results
 including dives, energy expenditure and success rates can be seen in Table 4.

**Figure 3.** Minimum feeding success rates between the sexes to maintain body condition and
feed a chick. Males were predicted to require a higher feeding success rate due to the lower
numbers of dives undertaken.

Bird ID	Sex	Year of study	Tracking duration (Days)	Number of dives	Dives per day	Total energy expenditure during tracking (kJ)	Total energy expenditure during tracking plus chick demands (kJ)	Energy expenditure per day with chick demands (kJ)	Modelled average kJ per successful dive	Number of successful dives to meet energy demands	Percent of recorded dives needed to be successful
D01	Male	2017	4.90	113	23.06	19017.89	21756.33	4440	1005.04	21.65	19.16
D02	Male	2017	2.86	36	12.56	9977.19	11577.81	4042	1005.04	11.52	31.99
D03	Unknown	2017	5.08	189	37.15	17676.91	21941.57	4313	NA	NA	NA
D04	Female	2017	0.97	65	66.89	3728.53	4543.13	4675	1005.96	4.52	6.94
D05	Female	2017	1.83	39	21.31	6820.18	8354.34	4565	1005.96	8.30	21.29
D12	Female	2017	4.68	90	19.19	18037.97	21968.74	4685	1005.96	21.84	24.27
D13	Male	2017	4.72	87	18.44	16414.58	19050.09	4040	1005.04	18.95	21.79
D16	Male	2017	1.99	18	9.06	6742.38	7852.85	3952	1005.04	7.81	43.41
D25	Female	2018	3.04	37	12.17	11665.59	14212.90	4677	1563.34	9.09	24.57
D26	Male	2018	2.92	36	12.31	12383.63	14017.50	4794	1552.61	9.02	25.08
D28	Male	2018	4.61	230	49.84	16418.62	18997.12	4117	1552.61	12.24	5.32
D41	Male	2018	4.83	39	8.07	17065.3	19766.91	4089	1552.61	12.73	32.64
D52	Female	2018	4.32	42	9.72	15395.66	19016.77	4402	1563.34	12.16	28.96
D53	Male	2018	5.05	25	4.95	18274.86	21095.85	4179	1552.61	13.59	54.35

**Table 4.** Summary of results from tracked birds between 2017 and 2018. Energy expenditure is calculated from the formulae in figure 2 and
chick demands are included by the amount of energy required by a four-week-old chick. Modelled average kJ per successful dive includes results
from a Bayesian mixed model from isotope analysis and is produced as a figure for each sex per year.

Discussion

Here we show that, for gannets, sex differences in foraging behaviour are not the result of
divergent energetic costs of foraging or different energetic content of consumed prey. We 373 suggest that sex differences in foraging behaviour are likely to have arisen from unequal
energetic demands between the sexes coupled with resource partitioning to avoid intraspecific
competition. SIA indicated sex-specific diets, but there was no difference in energy intake
between the sexes. Cost of individual prey capture attempts associated with differing diets was
low compared to total energetic expenditure, and despite females diving more than males, there
was no difference in energetic expenditure per day between the sexes.

To the best of our knowledge, this is the first time that the energetic cost of individual prey
capture attempts has been estimated in seabirds. Dynamic body acceleration is an established
proxy measure of energy expenditure (74), though difficulties remain in converting DBA to a true
measure of energy expenditure (26). Studies comparing DBA with energy expenditure must
ensure that summed values of energy expenditure must not be regressed against sum values of
DBA through time, a problem known as the time trap (75, 76). In this study we accounted for
time, allowing for meaningful estimates of energy expenditure per unit time from DBA. Our
method also bypasses the problem of changing metabolic rates, as we studied the cost of
behaviour and locomotion only. The relationship between kilojoules and time intersects at 0,
therefore avoiding the need to calibrate acceleration to metabolic rate (77). Though we do not
account for the error of environmental influences, we have assumed that this variance is equal
between individuals. The resulting energetic cost of prey capture events was low, even after
including the cost of take-off from the sea surface following a dive, with all prey capture attempts
across a foraging trip accounting for <4% of total energy expenditure. This suggests that the cost
of diving probably does not limit the number of prey capture attempts in gannets, though we
acknowledge this may not be true for birds struggling to meet daily energy demands. Sex
differences in the energetic cost of individual prey capture attempts were minor and non-
significant, albeit based on a small sample size. Despite females undertaking an average of eight
more dives per day, the low cost of prey capture attempts resulted in no differences in daily

energetic expenditure between males and females. Females diving more may expend relatively
more energy as they spend more time underwater, however this is likely not the case as it has
been found that metabolic energy expenditure is not affected by the medium an animal moves
through (78). Year and dive type (plunge versus pursuit dive) had the largest effect on energetic
cost of diving, reflecting yearly differences in diet noted in SIA analysis, that are likely related to
the proportion of different dive types. 76% of dives were plunge dives with an almost immediate
rise to the surface, though 2017 contained 12.9% more pursuit dives than 2018. The increased
cost of underwater pursuit following a 'failed' plunge dive suggests a cost-benefit trade-off, and
Machovsky-Capuska, Vaughn (79) noted higher feeding success in pursuit dives in Australasian
gannets, *Morus serrator*, that would support this hypothesis.

Gannets forage on a wide variety of prey (80), and SIA models indicated divergent diets between
males and females, consistent with previous studies in gannets (42, 46). Prey proportions from
our SIA models were similar to those previously reported by Lewis, Sherratt (61) at the same site,
and we found females took proportionately more mackerel and less whiting, Norway pout, and
herring compared to males. Applying the average calorific content of prey species to sex-specific
diets, energetic gain per dive did not differ between sexes. However, females make a greater
contribution to chick provisioning (71), which may require a proportionate increase in targeting
of smaller sized prey for chick consumption. While this has been observed in other seabird
species (81), there is little evidence to suggest such specific prey targeting in provisioning gannets
whose chicks are capable of consuming quite large prey items. Our results support the suggestion
that divergent diet is not the result of differing energetic cost of prey capture, or energy content
of prey but may be a result of intersexual competition, as previously demonstrated in this
population of gannets (46).

Intraspecific competition is expected to be higher with increasing proximity to a breeding colony
(82, 83) and this competition may drive differing sexually divergent foraging behaviour in
gannets. Several studies report that male gannets forage closer to breeding colonies whilst
females travel further (42, 44). This may be due to male gannets outcompeting females forcing
them to travel further and undertaking different dive behaviour as they are forced to forage in

different habitat than males (45, 46). However, these studies concede that there is no strong
compelling evidence that sexual separation is entirely due to males outcompeting females.

Different nutritional requirements between the sexes may also drive divergent foraging
behaviour. As the sexes search for different prey, they may engage in alternative prey capture
and foraging behaviour. One component of birds' life history strategy that may induce specific
nutrition demands is egg production. Egg production by females may cause a nutrient deficit (84,
85), specifically of calcium (86), which may drive different foraging behaviour as birds seek to
recover this loss (87). Gannets are known to lay small eggs in comparison to their body size (88)
so it is currently unknown how this may affect foraging requirements.

Female gannets dived more frequently than males which may reflect differing provisioning roles
(89, 90), with female gannets estimated to have a 9.6% higher daily energetic demand, largely
because of their greater contribution to chick feeding (Montevicchi et al., 1984). After
accounting for the increased energetic demands in females, the energetic cost of foraging, the
mean calorific content of prey in sex-specific diets, and the number of dives performed, males
were predicted to have a higher minimum feeding success rate than females (21% of dives in
females and 29% of dives in males). These estimates of feeding success are lower than previous
estimates of approximately 50-66% for Australasian gannets based on identifying prey captures
from bird-borne cameras (91, 92). Our estimates reflect *minimum* success rates required to meet
energy demands, and the discrepancy suggests that gannets may routinely catch more food than
required to meet minimum energy demands that may be invested in chick provisioning, or that
they engage in energetically demanding activities around and within the colony such as preening
and aggression (88) that are not accounted for in our analysis.

Energy acquisition and allocation provide a useful framework to study ecological problems,
including management and evolution (93). This study highlights how DBA can estimate energetic
cost of discrete behaviours as well as overall energetic expenditure across defined time periods,
providing insights into the foraging ecology of free-ranging animals. While gannets are sexually
monomorphic, they show divergent foraging behaviour and diet, which our results suggest are
not the result of differing cost of foraging or energy content of prey. Such sexually divergent

foraging strategies in monomorphic species are thought to be driven by intersexual competition
or differing energy demands (20). In gannets, sex differences in foraging might be driven by a
combination of both processes; intersexual competition (46) and higher energetic demands in
females due to unequal chick provisioning. Female gannets meet this additional need through
increased dive rate, a strategy that has no appreciable additional cost given the small overall cost
of individual dives and may be an adapted strategy to account for competitive exclusion. Over
the course of a breeding season, this extra energetic expenditure equates to approximately 1567
461 kJ, less than the energy provided by one mackerel. However, after accounting for the cost of
462 dives, the energetic content of prey, and the number of dives performed, females appear to have
463 lower overall success rates to meet energetic requirements, suggesting some subtle difference
in foraging behaviour that remains unknown.

Our methodology and results have highlighted that in northern gannets, a sexually monomorphic
species, the sexes show differences in foraging behaviour primarily related to dive rate and
feeding success rather than the energetic cost of foraging. Evaluating sex differences in foraging
behaviour from an energetic perspective may provide a clearer picture for understanding
sexually divergent foraging strategies in both sexually monomorphic and dimorphic species.
Future research should consider an energetics approach in exploring the fine scale behavioural
differences between sexes.

**References**

- 1. Ginnett TF, Demment MW. Sex differences in giraffe foraging behavior at two spatial scales.
*Oecologia*. 1997;110(2):291-300.
 - 2. Beck CA, Bowen WD, McMillan JI, Iverson SJ. Sex differences in diving at multiple temporal
scales in a size - dimorphic capital breeder. *Journal of Animal Ecology*. 2003;72(6):979-93.
 - 3. Phillips R, Silk J, Phalan B, Catry P, Croxall J. Seasonal sexual segregation in two *Thalassarche*
albatross species: competitive exclusion, reproductive role specialization or foraging niche divergence?
*Proceedings of the Royal Society of London Series B: Biological Sciences*. 2004;271(1545):1283-91.
 - 4. Patrick SC, Weimerskirch H. Consistency pays: sex differences and fitness consequences of
behavioural specialization in a wide-ranging seabird. *Biology letters*. 2014;10(10):20140630.
 - 5. Wanless S, Harris M, Morris J. Factors affecting daily activity budgets of South Georgian shags
during chick rearing at Bird Island, South Georgia. *The Condor*. 1995;97(2):550-8.
 - 6. Maklakov AA, Simpson SJ, Zajitschek F, Hall MD, Dessimann J, Clissold F, et al. Sex-specific fitness
effects of nutrient intake on reproduction and lifespan. *Current Biology*. 2008;18(14):1062-6.

7. Reddiex AJ, Gosden TP, Bonduriansky R, Chenoweth SF. Sex-specific fitness consequences of
nutrient intake and the evolvability of diet preferences. *The American Naturalist*. 2013;182(1):91-102.
- 8. Kokko H, Jennions MD. Sex differences in parental care. *The Evolution Of Parental Care*. 484:
Oxford University Press; 2012.
- 9. Rogowitz GL, Chappell MA. Energy metabolism of eucalyptus-boring beetles at rest and during
locomotion: gender makes a difference. *Journal of Experimental Biology*. 2000;203(7):1131-9.
- 10. Lees JJ, Nudds RL, Folkow LP, Stokkan K-A, Codd JR. Understanding sex differences in the cost of
terrestrial locomotion. *Proceedings of the Royal Society B: Biological Sciences*. 2011;279(1729):826-32.
- 11. Biggerstaff MT, Lashley MA, Chitwood MC, Moorman CE, DePerno CS. Sexual segregation of
forage patch use: support for the social-factors and predation hypotheses. *Behavioural processes*.
2017;136:36-42.
- 12. Galezo AA, Krzyszczyk E, Mann J. Sexual segregation in Indo-Pacific bottlenose dolphins is driven
by female avoidance of males. *Behavioral Ecology*. 2017;29(2):377-86.
- 13. Pincheira - Donoso D, Tregenza T, Butlin RK, Hodgson DJ. Sexes and species as rival units of
niche saturation during community assembly. *Global Ecology and Biogeography*. 2018;27(5):593-603.
- 14. González - Solís J, Croxall JP, Wood AG. Sexual dimorphism and sexual segregation in foraging
strategies of northern giant petrels, *Macronectes halli*, during incubation. *Oikos*. 2000;90(2):390-8.
- 15. Ruckstuhl K, Neuhaus P. Sexual segregation in ungulates: a new approach. *Behaviour*.
2000;137(3):361-77.
- 16. Salton M, Kirkwood R, Slip D, Harcourt R. Mechanisms for sex-based segregation in foraging
behaviour by a polygynous marine carnivore. *Marine Ecology Progress Series*. 2019;624:213-26.
- 17. Ehl S, Hostert K, Korsch J, Gros P, Schmitt T. Sexual dimorphism in the alpine butterflies *Boloria*
*pales* and *Boloria napaea*: differences in movement and foraging behavior (*Lepidoptera: Nymphalidae*).
*Insect Science*. 2018;25(6):1089-101.
- 18. Tina F, Jaroensutasinee M, Jaroensutasinee K. Effects of sexual dimorphism and body size on
feeding behaviour of the fiddler crab, *Uca bengali* Crane, 1975. *Crustaceana*. 2015;88(2):231-42.
- 19. Wearmouth VJ, Sims DW. Sexual segregation in marine fish, reptiles, birds and mammals:
behaviour patterns, mechanisms and conservation implications. *Advances in marine biology*.
2008;54:107-70.
- 20. Pinet P, Jaquemet S, Phillips RA, Le Corre M. Sex-specific foraging strategies throughout the
breeding season in a tropical, sexually monomorphic small petrel. *Animal Behaviour*. 2012;83(4):979-89.
- 21. Elliott K, Gaston A, Crump D. Sex-specific behavior by a monomorphic seabird represents risk
partitioning. *Behavioural Ecology*. 2010;21:1024 - 32.
- 22. Smith EAE, Newsome SD, Estes JA, Tinker MT. The cost of reproduction: differential resource
specialization in female and male California sea otters. *Oecologia*. 2015;178(1):17-29.
- 23. MacArthur RH, Pianka ER. On optimal use of a patchy environment. *The American Naturalist*.
1966;100(916):603-9.
- 24. Tullock G. The coal tit as a careful shopper. *The American Naturalist*. 1971;105(941):77-80.
- 25. Pulliam HR. On the theory of optimal diets. *The American Naturalist*. 1974;108(959):59-74.
- 26. Wilson RP, Börger L, Holton MD, Scantlebury DM, Gómez - Laich A, Quintana F, et al. Estimates
for energy expenditure in free - living animals using acceleration proxies; a reappraisal. *Journal of*
*Animal Ecology*. 2019;89:161-72.
- 27. Brown DD, Kays R, Wikelski M, Wilson R, Klimley AP. Observing the unwatchable through
acceleration logging of animal behavior. *Animal Biotelemetry*. 2013;1(1):20.
- 28. Butler PJ, Green JA, Boyd I, Speakman J. Measuring metabolic rate in the field: the pros and cons
of the doubly labelled water and heart rate methods. *Functional ecology*. 2004;18(2):168-83.

29. Green JA. The heart rate method for estimating metabolic rate: review and recommendations.
Comparative Biochemistry and Physiology Part A: Molecular & Integrative Physiology. 2011;158(3):287-
304.
30. Frappell P, Blevin H, Baudinette R. Understanding respirometry chambers: what goes in must
come out. Journal of Theoretical Biology. 1989;138(4):479-94.
31. Shepard EL, Wilson RP, Quintana F, Laich AG, Liebsch N, Albareda DA, et al. Identification of
animal movement patterns using tri-axial accelerometry. Endangered Species Research. 2008;10:47-60.
32. Gleiss AC, Wilson RP, Shepard EL. Making overall dynamic body acceleration work: on the theory
of acceleration as a proxy for energy expenditure. Methods in Ecology and Evolution. 2011;2(1):23-33.
33. Halsey L, Shepard E, Quintana F, Laich AG, Green J, Wilson R. The relationship between oxygen
consumption and body acceleration in a range of species. Comparative Biochemistry and Physiology Part
544 A: Molecular & Integrative Physiology. 2009;152(2):197-202.
34. Halsey LG, Shepard EL, Wilson RP. Assessing the development and application of the
accelerometry technique for estimating energy expenditure. Comparative Biochemistry and Physiology
Part A: Molecular & Integrative Physiology. 2011;158(3):305-14.
35. Goldsworthy B, Young MJ, Seddon PJ, van Heezik Y. Stomach flushing does not affect apparent
adult survival, chick hatching, or fledging success in yellow-eyed penguins (*Megadyptes antipodes*).
Biological Conservation. 2016;196:115-23.
36. Thiebault A, Semeria M, Lett C, Tremblay Y. How to capture fish in a school? Effect of successive
predator attacks on seabird feeding success. Journal of Animal Ecology. 2016;85(1):157-67.
37. Hobson KA. Trophic relationships among high Arctic seabirds: insights from tissue-dependent
stable-isotope models. Marine Ecology Progress Series. 1993;95:7-.
38. Hobson KA, Piatt JF, Pitocchelli J. Using stable isotopes to determine seabird trophic
relationships. Journal Of Animal Ecology. 1994:786-98.
39. Bond AL, Jones IL. A practical introduction to stable-isotope analysis for seabird biologists:
approaches, cautions and caveats. Marine Ornithology. 2009;37(3):183-8.
40. Stock BC, Semmens BX. Unifying error structures in commonly used biotracer mixing models.
Ecology. 2016;97(10):2562-9.
41. Deakin Z, Hamer KC, Sherley RB, Bearhop S, Bodey TW, Clark BL, et al. Sex differences in
migration and demography of a wide-ranging seabird, the northern gannet. Marine Ecology Progress
Series. 2019;622:191-201.
42. Stauss C, Bearhop S, Bodey T, Garthe S, Gunn C, Grecian W, et al. Sex-specific foraging behaviour
in northern gannets, *Morus bassanus*; incidence and implications. Marine Ecology Progress Series.
2012;457:151-62.
43. Malvat Z, Lynch S, Bennison A, Jessopp M. Evidence of links between haematological condition
and foraging behaviour in northern gannets (*Morus bassanus*). Royal Society Open Science.
2020;7(5):192164.
44. Lewis S, Benvenuti S, Dall–Antonia L, Griffiths R, Money L, Sherratt T, et al. Sex-specific foraging
behaviour in a monomorphic seabird. Proceedings of the Royal Society of London B: Biological Sciences.
2002;269(1501):1687-93.
45. Cleasby IR, Wakefield ED, Bodey TW, Davies RD, Patrick SC, Newton J, et al. Sexual segregation in
a wide-ranging marine predator is a consequence of habitat selection. Marine Ecology Progress Series.
2015;518:1-12.
46. Giménez J, Arneill GE, Bennison A, Pirota E, Gerritsen HD, Bodey TW, et al. Sexual mismatch
between vessel-associated foraging and discard consumption in a marine top predator. Frontiers in
Marine Science. 2021;8:220.

47. Bodey TW, Cleasby IR, Votier SC, Hamer KC, Newton J, Patrick SC, et al. Frequency and
consequences of individual dietary specialisation in a wide-ranging marine predator, the northern
gannet. *Marine Ecology Progress Series*. 2018;604:251-62.
- 48. Vandenabeele SP, Shepard EL, Grogan A, Wilson RP. When three per cent may not be three per
cent; device-equipped seabirds experience variable flight constraints. *Mar Biol*. 2012;159(1):1-14.
- 49. Griffiths R, Double MC, Orr K, Dawson RJ. A DNA test to sex most birds. *Molecular Ecology*.
1998;7(8):1071-5.
- 50. Collins PM, Green JA, Warwick - Evans V, Dodd S, Shaw PJ, Arnould JP, et al. Interpreting
behaviors from accelerometry: a method combining simplicity and objectivity. *Ecology And Evolution*.
2015;5(20):4642-54.
- 51. Shepard EL, Wilson RP, Halsey LG, Quintana F, Laich AG, Gleiss AC, et al. Derivation of body
motion via appropriate smoothing of acceleration data. *Aquatic Biology*. 2008;4(3):235-41.
- 52. Wilson RP, White CR, Quintana F, Halsey LG, Liebsch N, Martin GR, et al. Moving towards
acceleration for estimates of activity - specific metabolic rate in free - living animals: the case of the
cormorant. *Journal of Animal Ecology*. 2006;75(5):1081-90.
- 53. Patterson A, Gilchrist HG, Chivers L, Hatch S, Elliott K. A comparison of techniques for classifying
behavior from accelerometers for two species of seabird. *Ecology and Evolution*. 2019;9(6):3030-45.
- 54. Qasem L, Cardew A, Wilson A, Griffiths I, Halsey LG, Shepard EL, et al. Tri-axial dynamic
acceleration as a proxy for animal energy expenditure; should we be summing values or calculating the
vector? *PloS ONE*. 2012;7(2):e31187.
- 55. Schmidt-Nielsen K, Knut S-N. *Scaling: why is animal size so important?: Cambridge university
press; 1984.*
- 56. Hicks O, Burthe S, Daunt F, Butler A, Bishop C, Green JA. Validating accelerometry estimates of
energy expenditure across behaviours using heart rate data in a free-living seabird. *Journal of
Experimental Biology*. 2017;220(10):1875-81.
- 57. Dunn RE, White CR, Green JA. A model to estimate seabird field metabolic rates. *Biology Letters*.
2018;14(6):20180190.
- 58. Jennings S, Cogan S. Nitrogen and carbon stable isotope variation in northeast Atlantic fishes
and squids: *Ecological Archives E096 - 226*. *Ecology*. 2015;96(9):2568-.
- 59. Logan JM, Jardine TD, Miller TJ, Bunn SE, Cunjak RA, Lutcavage ME. Lipid corrections in carbon
and nitrogen stable isotope analyses: comparison of chemical extraction and modelling methods.
*Journal of Animal Ecology*. 2008;77(4):838-46.
- 60. Phillips DL, Inger R, Bearhop S, Jackson AL, Moore JW, Parnell AC, et al. Best practices for use of
stable isotope mixing models in food-web studies. *Canadian Journal of Zoology*. 2014;92(10):823-35.
- 61. Lewis S, Sherratt TN, Hamer KC, Harris MP, Wanless S. Contrasting diet quality of northern
gannets, *Morus bassanus*, at two colonies. *Ardea*. 2003;91(2):167-76.
- 62. Hobson KA, Clark RG. Assessing avian diets using stable isotopes II: factors influencing diet-tissue
fractionation. *The Condor*. 1992;94(1):189-97.
- 63. Bearhop S, Waldron S, Votier SC, Furness RW. Factors that influence assimilation rates and
fractionation of nitrogen and carbon stable isotopes in avian blood and feathers. *Physiological And
Biochemical Zoology*. 2002;75(5):451-8.
- 64. Forero MG, Tella JL, Hobson KA, Bertellotti M, Blanco G. Conspecific food competition explains
variability in colony size: a test in Magellanic penguins. *Ecology*. 2002;83(12):3466-75.
- 65. Cherel Y, Hobson KA, Hassani S. Isotopic discrimination between food and blood and feathers of
captive penguins: implications for dietary studies in the wild. *Physiological And Biochemical Zoology*.
2005;78(1):106-15.
- 66. Gelman A, Carlin JB, Stern HS, Dunson DB, Vehtari A, Rubin DB. *Bayesian Data Analysis:
Chapman and Hall/CRC; 2013.*

67. Semmens BX, Stock B, Ward E, Moore JW, Parnell A, Jackson AL, et al. MixSIAR: A Bayesian
stable isotope mixing model for characterizing intrapopulation niche variation. Ecological Society of
America, Minneapolis, MN. 2013:04-9.
- 68. Vehtari A, Gelman A, Gabry J. Practical Bayesian model evaluation using leave-one-out cross-
validation and WAIC. *Statistics And Computing*. 2017;27(5):1413-32.
- 69. Barton K. MuMIn: Multi-model inference. R package version 1.0.0. Vienna, Austria: R
Foundation for Statistical Computing See [http://CRAN](http://CRAN.R-project.org/package=MuMIn)
R-project org/package= MuMIn. 2011.
- 70. Harrison XA, Donaldson L, Correa-Cano ME, Evans J, Fisher DN, Goodwin CE, et al. A brief
introduction to mixed effects modelling and multi-model inference in ecology. *PeerJ*. 2018;6:e4794.
- 71. Montevecchi WA, Ricklefs R, Kirkham I, Gabaldon D. Growth energetics of nestling northern
gannets (*Sula bassanus*). *The Auk*. 1984;101(2):334-41.
- 72. Montevecchi W, Kirkham I, Purchase R, Harvey B. Colonies of Northern Gannets in
Newfoundland. *Osprey*. 1980;11:2-8.
- 73. Cooper J. Energetic requirements for growth and maintenance of the Cape gannet (*Aves*;
*Sulidae*). *African Zoology*. 1978;13(2):305-17.
- 74. Elliott K, Le Vaillant M, Kato A, Speakman J, Ropert-Coudert Y. Accelerometry predicts daily
energy expenditure in a bird with high activity levels. *Biology Letters*. 2013;9:20120919.
- 75. Halsey LG. Relationships grow with time: a note of caution about energy expenditure - proxy
correlations, focussing on accelerometry as an example. *Functional Ecology*. 2017;31(6):1176-83.
- 76. Ladds MA, Rosen DA, Slip DJ, Harcourt RG. Proxies of energy expenditure for marine mammals:
an experimental test of "the time trap". *Scientific Reports*. 2017;7(1):1-10.
- 77. Halsey LG, Bryce CM. Proxy problems: why a calibration is essential for interpreting quantified
changes in energy expenditure from biologging data. *Functional Ecology*. 2021;35(3):627-34.
- 78. Laich AG, Wilson RP, Gleiss AC, Shepard EL, Quintana F. Use of overall dynamic body
acceleration for estimating energy expenditure in cormorants: does locomotion in different media affect
relationships? *Journal of Experimental Marine Biology and Ecology*. 2011;399(2):151-5.
- 79. Machovsky-Capuska GE, Vaughn RL, Würsig B, Katzir G, Raubenheimer D. Dive strategies and
foraging effort in the Australasian gannet, *Morus serrator*, revealed by underwater videography. *Marine*
*Ecology Progress Series*. 2011;442:255-61.
- 80. Hamer K, Humphreys E, Garthe S, Hennicke J, Peters G, Grémillet D, et al. Annual variation in
diets, feeding locations and foraging behaviour of gannets in the North Sea: flexibility, consistency and
constraint. *Marine Ecology Progress Series*. 2007;338:295-305.
- 81. Davoren GK, Burger AE. Differences in prey selection and behaviour during self-feeding and
chick provisioning in rhinoceros auklets. *Animal Behaviour*. 1999;58(4):853-63.
- 82. Ashmole NP. Seabird ecology and the marine environment. *Avian Biology*. 1971;1:223-86.
- 83. Wakefield ED, Bodey TW, Bearhop S, Blackburn J, Colhoun K, Davies R, et al. Space Partitioning
Without Territoriality in Gannets. *Science*. 2013;341(6141):68-70.
- 84. Scanes CG, Campbell R, Griminger P. Control of energy balance during egg production in the
laying hen. *The Journal of Nutrition*. 1987;117(3):605-11.
- 85. Perrins C. Eggs, egg formation and the timing of breeding. *Ibis*. 1996;138(1):2-15.
- 86. Keshavarz K. The effect of dietary levels of calcium and phosphorus on performance and
retention of these nutrients by laying hens. *Poultry Science*. 1986;65(1):114-21.
- 87. Reynolds SJ, Perrins CM. Dietary calcium availability and reproduction in birds. *Current*
*Ornithology Volume 17: Springer*; 2010. p. 31-74.
- 88. Nelson B. *The Atlantic Gannet*: Fenix Books; 2002.
- 89. Grieco F. Short-term regulation of food-provisioning rate and effect on prey size in blue tits,
*Parus caeruleus*. *Animal Behaviour*. 2001;62(1):107-16.

90. Limmer B, Becker PH. Improvement in chick provisioning with parental experience in a seabird.
Animal Behaviour. 2009;77(5):1095-101.
91. Cansse T, Fauchet L, Wells M, Arnould J. Factors influencing prey capture success and
profitability in Australasian gannets (*Morus serrator*). Biology Open. 2020;9:1.
92. Wells MR, Angel LP, Arnould JP. Habitat-specific foraging strategies in Australasian gannets.
Biology Open. 2016;5(7):921-7.
93. Karasov W. Energetics, physiology and vertebrate ecology. Trends in Ecology & Evolution.
1986;1(4):101-4.

Appendix B

Dear Editor,

Please find attached our resubmitted manuscript '*A bioenergetics approach to understanding sex differences in the foraging behaviour of a sexually monomorphic species*' for consideration by Open Science. We were very pleased at the positive responses of associate editors and reviewers, and have taken on board their comments and suggestions as outlined below. Additionally, elements of this work were presented at the World Seabird Conference, during which Dr Jonathan Green, a noted expert in seabird energetics, provided some excellent feedback. We have since liaised with him to incorporate some further refinements in addition to those of the reviewers, mostly centred on providing more details on the methodology. We hope you agree that the changes we have made have improved the manuscript and that it is now suitable for publication.

Thanks to his input we have now included Dr Green as an author on the paper, all current authors agree with this addition, we feel his additions have improved the manuscript and his inputs warrant authorship.

Best regards,

Ashley Bennison, on behalf of co-authors

Associate Editor Comments to Author (Dr Agustina Gómez-Laich):

In this study, the authors used a bioenergetic approach to examine intersexual differences in the foraging behavior of a sexually monomorphic seabird; the Northern Gannet. To do this, they instrumented female and male breeding gannets with GPSs, accelerometers, and TDRs. Energy expenditure was estimated using dynamic body acceleration and afterward converted to kilojoules. Additionally, Stable Isotope Analyses was employed to study intersexual differences in diet and to estimate each sex's energy acquisition. The study's main finding is that sex differences in foraging behavior are mainly associated with dive rate and success. This study provides an important contribution to science, but the methods and discussion sections need to be improved. I feel authors should address some general and specific issues (see below) before the manuscript can be published.

Authors response: Thank you for your review – we have reworked much of the methodology and discussion to try and make things clearer. Additionally, this work has since been presented at the World Seabird Conference, during which Dr Jonathan Green provided some excellent feedback which has helped to correct methods. We hope the changes we have made throughout the manuscript are appropriate.

General comments

1) Please present in a more explicit way the hypothesis you are testing. This would help to structure the manuscript. Additionally, it would greatly improve the manuscript

if authors refer to the main questions and hypotheses/predictions along with the methodology, results, and discussion sections. For example, several sections of the discussion refer to the methodology employed to estimate energy expenditure. Even though this is an important aspect of the Ms, it is not one of the main objectives.

Authors response: we have now included three specific hypotheses to tie the manuscript together better. These can be found on line 204-210 and are:

“

- 1) Sex differences in the foraging ecology of gannets derive from the different energetic demands placed upon the sexes.
- 2) Being a monomorphic species, there will be no difference in the cost of prey capture attempts between the sexes.
- 3) Due to differing energy demands and foraging behaviour, the sexes will have different prey capture success rates.

“

2) Several sections of the methodology need to be better explained (see below).

Authors response: [All sections of the methodology have been refined providing more detailed explanations as set out in response to specific points below.

3) Results could be improved by incorporating more tables for example as Supplementary material.

Authors response: The supplementary material has been expanded to include 5 tables and 2 figures – with explanation.

4) The discussion could be greatly improved if authors focused more on debating their findings and how they contribute to the main hypothesis. Many paragraphs repeat information that was already mentioned in the results.

Authors response: We have amended the discussion to better relate the findings to the hypotheses tested and in the context of other studies.

Specific comments

Keywords: accelerometry instead of accelerometry

Keywords: Isotope or isotope?

Authors response: Thank you – we have changed the keywords as suggested

Introduction

Lines 77-79. It would be interesting to explain how foraging at different locations may benefit females to restore their body condition after egg production.

Authors response: We have included a statement to describe this more effectively. This can now be found on line 93.

Lines 79-81. Can you give an example of how sex-specific foraging strategies in sexually monomorphic species may be driven by intraspecific competition causing one sex to be displaced spatially or to forage in different niches?

Authors response: Included example using brown boobies on line 99.

Lines 76-86. In this second paragraph, the first and fourth sentences present very similar ideas.

Authors response: To ease flow we have deleted the second sentence. This paragraph is now lines 92-103.

Line 102. Please give examples of other direct methods.

Authors response: We have included the direct methods of regurgitate sampling and direct observation of foraging. This is now on line 166:

“However, Stable Isotope Analysis (SIA) is a minimally invasive technique that can provide diet information and, in seabird studies, is known to correlate well with other more direct methods such as regurgitate sampling and direct observation of foraging”

Line 103. Can you briefly explain what each isotopic ratio allows us to know about prey consumption?

Authors response: This has been expanded and can be found on line 168-171:

“Both carbon and nitrogen can be considered as indicators of the trophic level an animal is foraging at (41). Nitrogen isotopes enrich at a faster rate in predators than carbon isotopes, but the ratio between them can inform trophic level, trophic niche width, and diet”

Line 107. Please add what this marginally heavier means.

Authors response: We have added a reference to show approximately 200g difference between the sexes which corresponds to approximately 6% of body mass. This is now on line 175:

“While females are marginally heavier than males (Approximately 200g, or 6% (45)), weight alone cannot be used to sex individuals”

Line 114-115. Than forage fish female specialist sounds awkward.

Authors response: This sentence has now been reworded so it reads better. It is now on line 182:

“Females that specialise on fisheries discards travel shorter distances than those who specialise on foraging for fish, although this distinction is not apparent among males”

Line 119-125. I suggest rephrasing this last paragraph. First, mention the objectives and then the methodology you will use to reach them. Additionally, It would be interesting to be more explicit with the hypothesis you are testing (see general comments). It would also be interesting to present some predictions.

Authors response: This has been rephrased and hypotheses included. This can now be found on line 197-210.

Methodology

Line 140. Figure 1 does not show the conceptual diagram of the study. This figure shows a diagram of a part of the methodology.

Authors response: This is has been changed to: “A visual diagram of the methodology is presented in figure 1.” This can now be found on line 240.

Line 140. How many males, and how many females were instrumented? Did you instrument pairs? How much time were the devices left on the birds? This information is mentioned in the results instead of being in the methodology. Did birds that were instrumented continued breeding normally?

Authors response: We have now included this information (five females and eight males) in the methods with time devices were on birds. We have also included a statement in this paragraph stating that we only deployed on a single parent within breeding pairs and that all pairs were observed continuing chick-rearing including feeding following the deployments. This paragraph can be found on line 240-263.

Line 141. This means that in some nests chicks were 21 days old and in other chicks were more than a month? Is the foraging behavior similar during both stages of the breeding season (3 vs 5 weeks old chicks)? Please incorporate information regarding this particular topic.

Authors response: Unfortunately it was not possible to work with chicks at the exact same ages, but tried to work with breeding birds that had chicks of similar age based on morphological descriptions. Gannets feed their chicks over a 12-13 week period before fledging, and the energy requirements of 3-4 week old chicks are largely similar (Montevecchi et al 1984), and so we considered this was appropriate.

Please add the dimensions of each one of the loggers you deployed on the birds.

Authors response: These have been added and can be seen on line 246-250: “GPS loggers (i-gotU GT-120, Mobile Action Technology Inc., Taipei, Taiwan, 14g, Dimensions: 4 x 2 x 1 cm)) recorded locations every 3 minutes; time depth recorders (TDR, CEFAS G5, 2.5g, Dimensions: 2 x 1 x 1 cm) recorded depth at 4Hz after exceeding a 0.5 or 1m depth threshold; tri-axial accelerometers (Gulf Coast Data Concepts X16-mini, 17g, Dimensions: 6 x 2 x 1 cm)”

Line 152. How much blood was taken? From where was the blood sample taken?

Authors response: Approximately between 1 and 1.5 ml were taken from the tarsus. This has been added to the methods on line 259:

“Between 1 and 1.5ml of blood was sampled from the tarsus vein for stable isotope analysis”

Figure 1. I suggest including which were the defined behaviors.

Authors response: This has now been changed to say dive behaviours. This can be seen on line 274.

Line 159-160. “further methods that develop the findings” sounds awkward. Please rephrase.

Authors response: This has been reworded and can be seen on line 279:

“Blue boxes represent the methodology for analysing data and orange boxes represent additional analysis.”

Line 162. Please explain more in detail how you analyze the acceleration data. To obtain a time-activity budget and estimate the energy expenditure during a foraging trip, not only the dives should be identified from the acc data. How did you recognize when birds were flying and floating? Equations to estimate energy expenditure from VeDBA may be activity-specific. For example, the equation used to derive energy expenditure for flying and diving may be different. Was this taken into account? Please give more details about this particular subject. Additionally, please give more details about how the acceleration data was processed. Once you had the acceleration data, you calculated the average value for each of the axes using a running mean of 2 seconds? How did you calculate the pitch values from the acc data? What does the X, Z, and Y-axis mean? Please specify which is the heave, sway and surge.

Authors response: We have tried to further explain the accelerometry process – we hope we have made the explanation clearer.

We have included the alternative name for each axis (surge, heave, and sway) as each one is first used but kept the namings of X-axis and Z-axis in the manuscript after this.

We have not undertaken a specific behavioural budget approach in this study. A large challenge with accelerometry data is understanding what signal relates to which specific behaviour. This is not possible on occasions where an animal is out of sight or is undertaking an unknown behaviour.

Furthermore, there are no equations to estimate energy expenditure directly from VeDBA for this (and the vast majority of other) species. Instead we adopt a new, simple, approach which allows estimation of the amount of energy used from VeDBA for specific periods of time or specific activities, for groups of animals. Firstly, we assume basal metabolic rate (BMR) and overall field metabolic rate (FMR) to be constant and defined only by each focal bird's mass (and latitude and species in the case of seabirds). This is

understandably a simplification but we hope we have addressed the consequences and limitations of our simple approach appropriately in the manuscript. Secondly, we use VeDBA scores to reflect energetic investment in a suite of behaviours, under a fundamental assumption that one unit of VeDBA is equivalent to a consistent amount of energy used, independent of time. A 24 hour period can contain many behaviours but the sum of VeDBA will reflect the costs of all of these between BMR and FMR. Based on this, the relationship between movement (VeDBA) and energy expenditure associated with periods of movement can be considered as linear. By combining these assumptions, we can start to estimate energetic costs of movement in different groups of the same species, in this case males and females. This is now explained in full in the manuscript.

Lines 172-175. It would be nice to see a figure of the bimodal distribution accelerometer-derived dives had as supplementary information.

Authors response: This has now been included in supplementary materials and has been mentioned in the main text on line 294.

Line 177. To do this first you had to obtain the time-activity budget of the birds, that is to say, how much time each bird expended on each activity. For this, the acceleration data such me labeled. How did you do this? Visually? Using some algorithm? Please give more details about this.

Authors response: We hope this has now been clarified in the text and explained above (comment for line 162).

Line 179. Please explain how you calculated VeDBA. It is not clear if you calculated the VeDBA for each activity (flying, floating, diving) and once this had been done using a specific equation to convert this VeDBA value into kilojoules. Which allometric equations did you use? It would be worth incorporating this as supplementary information also.

Authors response: We have provided more detailed information in the methods and hope this is now clearer. VeDBA was calculated across the tagging period, incorporating the entire range of behaviours and values converted to energy expenditure in kJ. This section has received considerable work and can be seen from line 298-365.

Line 191. What does the individualized VeDBA to kJ equation mean?

Authors response: The process for converting VeDBA to kJ uses an equation based on the difference between the estimates of resting metabolic rate and field metabolic rate. These estimates are more accurate when considering an animal's weight and so the process was undertaken for each gannet individually to produce unique gradients (slope k in figure 2.). The text has been changed to reflect this and can be seen on line 322.

Line 193. Couldn't you use the TDR information to determine when birds were on the surface?

Authors response: Unfortunately this was not a reliable method as the TDRs were set to activate by pressure (<0.5m depth) and so did not register wet/dry to determine rest periods on the water.

Line 204. n=19 in 2017 and n=28 in 2018 should be placed after 47 birds.

Authors response: This has been moved and can now be seen on line 257.

Line 234. Here it says LMER but in the results it says GLM. Did you perform an LMER or a GLM? Please add which distribution was used and why.

Authors response: This was an LMER. The result of the averaged LMER is presented at the beginning of the first section of the results entitled "Sex differences in dive behaviour." The GLM presented at the end of this section was not specifically mentioned in the methods and was an oversight. We have now included this in the methods section on line 433:

"The rates of dives per day between females and males was tested using a general linear regression, with dive rate as the response, predicted by sex as a factor. To determine if sex influences daily energy expenditure an LMER was used to predict energy expenditure (per day) from sex and year, with ID as a random effect to account for repeated measures from individuals."

Line 235. Please explain why the interaction between year and dive type was included in the model.

Authors response: The model did not include the interaction between year and dive type but did include the interaction between sex and dive type. This was included as females are slightly heavier, which may influence the cost of a dive. We have included a sentence to clarify this on line 429:

"The interaction between sex and dive type was included to explore if the different masses of the sexes (Approximately 200 g (45)) impacted the cost of a dive type"

Line 239. In general, there seems to be some controversy about model averaging. If your top model has relatively good support (as compared to second-best models) some suggested it may be better to refrain from model averaging. Why did you choose to do model averaging? Please explain how you obtained the average. It would be interesting to incorporate a table with the best models, their AIC, deltaic, and weight as supplementary material.

Authors response: We do agree that sometimes model averaging can be overused, however we consider that in this case it is appropriate. Supporting models were all relatively close in AIC and allowed for a more integrative approach to understanding the results from the perspective of sex differences.

We have included the table of other models in the supplementary materials.

Line 246. These sentences are not clear. It is not clear if you used trips as energetic units or you also considered some periods at the colony.

Authors response: This has been reworded and we hope it is clearer now. This paragraph has also been moved to line 331:

“As gannet foraging trips may last several days, they incur increasing energetic costs during a foraging trip such as feeding chicks upon return, we have included this in the analysis by considering energetic differences from a whole sum approach and use individual energy expenditure per 24-hour period as the energetic unit.”

Line 247. So you calculated a foraging trip energy expenditure and a 24 hour period energy expenditure? Please clarify this aspect. Comparisons between females and males were performed for both time periods (foraging trip and 24-hour period)?

Authors response (For queries on lines 246 and 247): 24 hour period was the energetic unit. We have clarified this in the text on line 336 (text provided above for line 246 query).

Line 254. Did you mean the total amount of food they needed to eat to get the energy they expended? Cant this be achieved by eating more than one combination of prey proportions?

Authors response: Yes, the Total Energetic Demand (TED) is the energy required to be captured by a gannet to meet the expenditure from behaviour and to feed a chick, assuming no change in body mass. This could be met by any combination of prey proportions, we have therefore used the average value of a successful prey catch from the isotopic modelling.

Line 267-268. Please rephrase this sentence. It is not clear.

Authors response: This has been reworded and is on line 440:

“The number of dives successful dives required was then considered as a proportion of the number of dives undertaken; therefore presenting a minimum percentage of dives which must have been successful for each individual gannet to survive.”

Results.

Line 282. Does this mean that from one individual you couldn't determine the sex from the blood sample?

Authors response: Yes, this has been clarified in the text on line 491:

“Of the 14 gannets tracked, five were female, eight were male, and one was of unknown sex. due to inconclusive DNA test”

Line 290. Please explain better how this average LMER was obtained. It would be nice to see in a table all the models that showed delta AIC values higher than 6.

Authors response: The model averaging was undertaken using the model averaging function in the MuMin package – we have included this in text (Line 432) and a table of models in the supplementary material.

Line 316. Why here you present a Chi-square and in Line 313 an F? How did you get these statistics? This is not mentioned in the methodology.

Authors response: Both models were tested against a null model in an ANOVA. Both models should have used an F test, as is appropriate for continuous data. This was an error and has been corrected and replaced with the F statistic.

Figure 3. Please explain what each part of the boxplot means.

Authors response: This has been included on the figure 3 caption

Discussion

Line 385-387. This sentence is not clear. I would rather say that you focused on behaviors that imply movement. Certain behaviors do not imply movement and for that behaviors, VEDBA would not be useful.

Authors response: This has been changed to include the emphasis on behaviours that imply movement and can be seen on line 732:
“as we studied the cost of behaviours that are implied by movement only”

Line 399-401. Energy expenditure can be affected by the medium in which an animal moves especially if the movement in different media involves different muscle groups.

Authors response: Thank you, this has been reworded to change emphasis of sentence and can be seen on line 766:

“Females diving more may expend relatively more energy as they spend more time underwater. However, energy expenditure can be affected by the medium an animal moves through (84) and this then may affect the sexes unevenly, though this is unlikely given the proportionally low energetic costs of diving.”

Line 449. It would be nice to see a table showing how much energy each bird expended in the different behaviors that comprise the foraging trip. In this table, the time engaged in each behavior could also be included.

Authors response: Unfortunately, we do not have this information. As noted above, not being able to directly observe birds means that we could not confidently relate specific behaviours to signals within accelerometry data. Furthermore, as we now describe in more detail, our energetics methodology did not require a complete time budget to estimate total energy costs.

Line 470-471. Can you please more information and specific examples about how

the bioenergetic approach presented in this study could contribute to future studies?

Authors response: This has been expanded on line 897:

“It would be interesting to see this study replicated using more obviously dimorphic species, where differences between the sexes are more clearly pronounced, or to examine how the sexes may differ in their energy expenditure with changing prey resources (96).”

Reviewer comments to Author:

Reviewer: 1

Comments to the Author(s)

I really enjoyed reading this manuscript! Well done. The introduction was great and flowed nicely, providing a background into sex-specific differences in foraging behaviour, its potential drivers (energy expenditure and diet), the methods used to investigate this and your study system. This set the manuscript up well.

Authors response: Thank you for your positive comments. We have actually made some further refinements based on reviewer comments and following feedback after presenting some of this research at the World Seabird Conference.

Are you able to provide any further information regarding the “marginal” sex differences in weight observed in gannets, that you mention within your introduction? Are the other sex differences in gannets that you mention (foraging behaviour and diet) considered to ubiquitous, or are they only true of particular populations from/foraging in particular locations?

Authors response: We have included more information and provided some extra detail including weight differences and what this equates to in terms of % body mass. This can be seen on line 175:

“While females are marginally heavier than males (Approximately 200g, or 6% (45)), weight alone cannot be used to sex individuals (45).”

In your methods section, your field methods are good and descriptive but could include a bit more detail with regards to the potential for loggers to have impacted the gannets’ behaviour/demographic parameters. Some would argue that total deployment weights as well as the weights of the birds (upon deployment and retrieval possibly) should be included.

Authors response: A table of deployment and retrieval weights has now been included in the supplementary materials. We have also included a statement regarding the potential for individual behaviour effects from tagging on line 242-246.

You also don’t currently provide any methodology behind logger retrievals, including how many birds/loggers were recaptured and when this occurred (i.e., how long the

deployment length was). I know that this is mentioned later within your results, but wonder whether it should be considered as more of a methodological point.

Authors response: This has now been included on line 242:

“Five female and eight male gannets were tagged over the two years. Birds were equipped with tags for an average of 3.70 ± 1.39 days. To reduce potential impact on a breeding pair, only one individual of a pair was tagged for this study.”

I also wonder whether you should move your bloods methods to the Data Collection section, as I was surprised to read this section of text later on within your methods instead. Otherwise, perhaps you could rename your section heading methods so that they read “Biologging Data Collection” instead, or something similar.

Authors response: We have now included more blood collection information in this section on line 256:

“Blood samples were taken from the tarsal vein of 47 birds ($n=19$ in 2017 and $n=28$ in 2018), including the accelerometer-equipped birds, and used to construct a population model of dietary intake from isotope analysis (See section “*Isotopic Analysis for Diet Composition*” below). Between 1 and 1.5ml of blood was sampled for stable isotope analysis (see below) and 2-3 breast feathers were plucked for genetic sexing following the method outlined by Griffiths, Double (51).”

I feel that Figure 1 is a conceptual diagram of your study methods, as opposed to the actual study and your specific hypotheses?

Authors response: This has been renamed as a conceptual diagram in the text.

Please can you clarify the methods behind “confirming” TDR dives? Was this via visual inspection, as suggested in your Results?

Authors response: This was done by visual inspection and detail has now been included in the text on line 292:

“to validate accelerometer-derived dive events by visually comparing timestamps to TDR confirmed dives, this required each dive to be manually viewed and checked to compare with a dive from a TDR.”

I wonder whether it might be helpful to rearrange the order of your “Energetics from Accelerometry” section so that you first state what you are aiming to do with these methods, and then outline the steps that you took to achieve this goal.

Authors response: This section has now received a rework and reordering of the text to make the method clearer this section can be seen from lines 298 to 365.

I wonder whether the second paragraph of your “Statistical Analysis” section should feature earlier on as I’m not sure that it is really describing statistical analyses particularly. Perhaps this is also true of some of the following paragraph, i.e., the fish allometry etc.

Authors response: The second paragraph has now been moved to the energetics from accelerometry section (Line 332 to 365) and much of the following paragraph is now under the isotope section (Line 416 to 424).

When discussing sex differences in diving behaviour within your Results, perhaps considering including the percentage differences between some of the male and female metrics within the results would be helpful, rather than just the means of each sex.

Authors response: Where appropriate we have now included this. This can be seen on line 500 and 506:

“Combined cost of a single prey capture attempt (dive + take-off) in females was 1.94 ± 0.65 kJ while for males it was 1.74 ± 0.83 kJ, suggesting that male dives are 11.2% less costly than females. An averaged LMER indicated a significant effect of dive type and year on energy expenditure associated with dives, while sex was retained as a non-significant factor (Table 1, and Table S4 for model averaging results). The estimates for cost of all prey capture attempts represent < 4% of the daily total energy expenditure for each individual (Table S3). Accounting for unequal provisioning of the chick, and the cost of foraging, daily energetic demands were 10.28% higher for females than males”

I think that your table and figure headings could be more descriptive so that they are able to be easily interpreted as stand-alone items, without the remainder of the text being read. For example, you could include the species and colony that you are investigating.

Authors response: Text in the captions has been updated where appropriate.

I'm not sure whether I agree that there are not previous instances of the energetic cost of individual prey capture attempts being estimated in seabirds. Haven't seabird-mounted cameras paired with accelerometry been used to do this? Or devices that record beak opening events/changes in oesophageal temperature? Maybe I'm wrong, but perhaps these are methods that could also be mentioned in your Introduction if trying to estimate the energetic cost of prey capture attempts is a key goal of this manuscript.

Authors response: That is correct – we have changed the text of the paragraph to reflect this. This paragraph is now on line 725.

I think that your Discussion could generally do with another check through to ensure that the readability is as good as elsewhere in your manuscript and that it flows and covers all of the aspects that you want it to, in a way that flows and makes sense.

Authors response: Thank you – we have edited throughout the document, particularly the discussion to ensure good flow.

For example, I'm not totally sure what the goal of the large second paragraph is at the moment as you discuss a number of different results in turn throughout.

Authors response: This paragraph has now been split into several paragraphs with more detailed discussion in each.

Additionally, I think that L413-20 in particular could be streamlined a little to increase their readability. I know what you're trying to say, but I think that they could benefit from a little more editing, including the mention of it being the sexes that have divergent diets within the final sentence of this paragraph.

Authors response: Thank you – we hope we have increased the readability here.

I've recommended some grammatical changes to L421-7 too, but also wonder whether you could tie this back to the results of this manuscript a bit more. The same is also true of the following paragraph (L428-34) and elsewhere within your Discussion.

Authors response: Thank you – we have incorporated how these may be reflected in our results. The section on egg production has been removed, as this is undertaken many weeks prior to our deployments and is unlikely to have impacted the observed behaviour.

Some of your in-text citations seem to be in a strange format and should be double checked throughout.

Authors response: Thank you – these have been checked and corrected.

I've provided a marked-up document of the pdf with a few more small comments here and there, but otherwise, good job!

Authors response: Thank you – the comments from the PDF are copied below for reference.

Further comments from PDF:

Line 51: "The energetic cost?"

Authors response: Inserted

Line 73: "Consider including "for example" so that you introduce why you're talking about giant petrels now when the paragraph is so broad

Authors response: Inserted

Line 77: "are"?

Authors response: Inserted

Line 78: "breeding season,"

Authors response: Inserted

Line 97: "the"

Authors response: Inserted

Line 107: Rephrase sentence

Authors response: This has been made clearer

Line 107: How marginal please?

Authors response: Approximately 200g or 6% body mass – the manuscript has been update to reflect this and provides a reference

Line 116: consider “for whether male and female gannets target different sized prey items”

Authors response: Inserted

Line 121: Please consider adding a mention of gannets in this paragraph

Authors response: Inserted

Line 170: Please consider “Data from a subset of birds...”

Authors response: Inserted

Line 216: Please clarify which authors

Authors response: Clarified and inserted a few more words to make sense.

Line 288: Delete word “increased”

Authors response: Deleted

Line 291: The sentence needs a little bit of work perhaps a comma after “dives”

Authors response: inserted comma – sentence flows better now

Line 372: Writing “we suggest instead” would tie these sentences together nicely

Authors response: Inserted

Line 412: “after applying...”?

Authors response: Sentence changed

Line 422: Remove differing

Authors response: Inserted

Line 424: “females, forcing”

Authors response: Inserted

Line 425: “undertake”

Authors response: Inserted

Line 426: “habitat to males”

Authors response: Inserted

Line 435: “which may be reflective of”

Authors response: Inserted

Line 436: “than male gannets”

Authors response: Inserted

Line 441: I don't know whether you need the word previously here because of the estimates being for different species

Authors response: Agreed and removed

Line 447: I'd consider this sentence and breaking it down into multiple smaller ones

Authors response: This sentence has been broken into smaller sentences and some rewording done to improve flow

Line 448: questions (or something similar) rather than problems?

Authors response: Changed to questions

Line 449: “can be used to estimate the energetic costs of”

Authors response: Inserted

Line 453: consider “instead, such sexually”

Authors response: Inserted

Line 458: “through increasing their dive rate/rate of diving”?

Authors response: Inserted

Appendix C

Dear Editor,

Please find attached our resubmitted manuscript '*A bioenergetics approach to understanding sex differences in the foraging behaviour of a sexually monomorphic species.*'

We were very happy to receive minor corrections after resubmitting this manuscript, having integrated comments from the review process. We have addressed all comments provided to us at this stage and we hope you agree that the changes we have made have improved the manuscript and that it is now suitable for publication. We have included two copies of the manuscript, one with tracked changes from the last review, and a second copy with no track changes. We hope this is suitable.

Best regards,

Ashley Bennison

Introduction.

Line 88. Please incorporate the Brown booby specific species name.

Authors response: This has now been included

Lines 137-144. I realize in the previous version my suggestion was to first state the objectives and afterwards the technology employed. In the present version I suggest first mentioning the main objective and methodology employed and afterwards mention the specific objectives. For example:

In the present study, we used GPS, accelerometry, and SIA data to gain a better understanding of how gannets engage in foraging and how different demands upon the sexes may affect foraging strategies. Specifically, we explore sex differences in foraging of gannets in terms of diet, dive types, frequency of prey capture attempts, and the energetic cost of prey capture attempts. Additionally, we quantify the energetic requirements of each sex, taking into account energy expended during foraging and, using data from published studies, energetic demands of feeding offspring. Finally, we consider minimum dive success rates necessary for male and female gannets to meet their energy demands.

Authors response: This has been reworded as requested – thank you for your suggestion.

Lines 151-157. Hypothesis. The first one is fine however, the second and the third one are predictions not hypotheses. Please rephrase them.

Authors response: The hypotheses have been reworded to reflect the change between prediction and hypothesis.

Methods

Line 174. Please revise the numbers, here the total number of instrumented birds is 14 and below is 13. Please state how many females and males were equipped each year.

Authors response: This has now been clarified the total of 13 did not account for the individual of unknown sex. This has now been clarified and states:

“In 2017; three female, four male, and one unknown gannets were tagged, four males and two females were then tagged in 2018.”

Line 175. Please change for 52° 7' 37.92" N, 6° 35' 45.6" W

Authors response: This has been changed.

Line 181. Which was the depth threshold? 0.5 or 1 m? Or some devices were programmed with a 0.5 threshold and some with a 1 m threshold? Please clarify this aspect.

Authors response: Devices were mixed in the programming – as we tried to make the best regime possible. This has now been clarified and states:

“after exceeding depth threshold of either 0.5m or 1m depending upon tag setup”

Line 189. Is a period missing after (52)? The following that starts with “Previous” sounds a bit awkward, please rephrase it.

Authors response: We were missing a period. Thank you it has now been inserted.

Line 189. the “s” in gannets looks like a subscript letter.

Authors response: This has been corrected and is the appropriate size again.

Line 192. This is the first time a table of the Supplementary information is mentioned so I suggest considering this table as table S1 instead of table S5. Please check that all Supp. table numbers are correctly mentioned in the main document after they are renumbered.

Line 268. For example, this would be table S2 now.

Authors response to lines 189 and 268: We have now checked and reordered the supplementary material so that supplementary materials appear in order.

Line 293-297. This sentence is too long and not clear. Please rephrase it.

Authors response: This sentence has been broken into three smaller sentences.

Line 336. Please check Supplementary information Table numbers.

Authors response: The supplementary tables have now been ordered appropriately.

Line 368. It is not clear to me how you test for differences in diving rate between sexes using linear regression. Please clarify this aspect.

Authors response: This has been clarified as a general linear model and that sex is a predictive factor with dive rate as a response variable.

Results.

Line 421. Please mention in the methods how you tested for differences in body mass between sexes.

Authors response: This has now been included on line 346.

Line 426. Why didn't you test for differences in dive duration between sexes?

Authors response: We have now included this as an unpaired t-test reporting no significant differences in dive length between males and females. We have also included a statement in the methods stating this would be done.

Line 428. Why didn't you test for differences in dive + take off costs between sexes?

Authors response: This was tested more formally as part of the averaged LMER that used sex as a factor in the model predicting cost of dive.

Line 461. In the methods, you mention that differences in the diving rate between sexes were tested by means of linear regression and here a GLM is mentioned. Please clarify this aspect.

Authors response: We have clarified in the methods that it is in fact a GLM.

Line 501. You can say KIV instead of "average energy intake (KIV)" since you have already defined what KIV stands for.

Authors response: This has been amended as suggested.

Discussion

Line 586. Please eliminate "do" from "Females may have to do dive more".

Authors response: This has been removed.

Line 596. "gannets" can be eliminated here since it is clear you are talking about gannets.

Authors response: This has now been removed